# A Bayesian Hierarchical Model for Glacial Dynamics Based on the Shallow Ice Approximation and its Evaluation Using Analytical Solutions

Giri Gopalan[1], Birgir Hrafnkelsson[1], Guðfinna Aðalgeirsdóttir[2], Alexander H. Jarosch[2], and Finnur Pálsson[2]

[1]Faculty of Physical Sciences, School of Engineering and Natural Sciences; University of Iceland
[2]Institute of Earth Sciences; University of Iceland

*Correspondence to:* Giri Gopalan (gopalan88@gmail.com)

**Abstract.** Bayesian hierarchical modeling can assist the study of glacial dynamics and ice flow properties. This approach will allow glaciologists to make fully probabilistic predictions for the thickness of a glacier at unobserved spatio-temporal coordinates, and it will also allow for the derivation of posterior probability distributions for key physical parameters such as ice viscosity and basal sliding. The goal of this paper is to develop a proof of concept for a Bayesian hierarchical model constructed, which uses exact analytical solutions for the shallow ice approximation (SIA) introduced by Bueler et al. (2005). A suite of test simulations utilizing these exact solutions suggests that this approach is able to adequately model numerical errors and produce useful physical parameter posterior distributions and predictions. A byproduct of the development of the Bayesian hierarchical model is the derivation of a novel finite difference method for solving the SIA partial differential equation (PDE). An additional novelty of this work is the correction of numerical errors induced through a numerical solution using a statistical model. This error correcting process models numerical errors that accumulate forward in time and spatial variation of numerical errors between the dome, interior, and margin of a glacier.

## 1 Introduction

The shallow ice approximation (SIA) is a nonlinear partial differential equation (PDE) that describes ice flow when glacier thickness is relatively small compared to the horizontal dimensions. Derived from the principle of mass conservation, the SIA PDE depends on two key physical parameters: ice viscosity and basal sliding (sometimes described as basal friction or drag). The primary objective of this paper is to develop a Bayesian hierarchical model (BHM) for glacier flow utilizing the framework espoused by Wikle (2016) and Cressie and Wikle (2015), which allows one to: 1) infer ice viscosity and basal sliding parameters and 2) make probabilistic predictions for glacial thickness at unobserved spatio-temporal coordinates. This BHM relies upon a finite difference scheme for solving the SIA that is inspired by the Lax-Wendroff method (Hudson). To validate this BHM, we utilize exact analytical solutions from Bueler et al. (2005). Hence, in addition to the development of a BHM for shallow glaciers, this paper serves as a case-study for the strategy of using exact analytical solutions to validate or tune BHMs governed by physical dynamics. Moreover, the BHM developed can be applied to the general "physical-statistical"

problem (Berliner, 2003). This BHM is verified and diagnosed through a combination of assessments of posterior probability intervals, checks of predictive accuracy for glacial thickness prediction, and a comparison between observed and expected errors due to the numerical solution of the SIA.

## 1.1 An Overview of Bayesian Modeling and BHMs

Before describing how BHMs are used in physical-statistical models, particularly for geophysical problems, a very terse overview of Bayesian modeling and Bayesian hierarchical modeling is given for the uninitiated reader. A main component of Bayesian statistics is the use of probability distributions to model parameters thought to be fixed quantities (i.e., scientific constants); this assumption allows one to use rules of conditional probability (i.e., Bayes' theorem) to derive probability distributions for scientific quantities of interest, such as physical constants or predictions of future quantities of a system being studied. Typically, the major assumptions required as input to the analysis are prior distributions for parameters as well as a probabilistic model for the data. The output is a probability distribution for parameters or predictions conditional on data that has been collected or observed; canonically, this is referred to as the posterior distribution.

A BHM is a Bayesian model in which the probabilistic model for data is specified in a hierarchy. Working with such a hierarchy has a number of advantages – it is usually easier to conceptualize the probabilistic model for the data, and it is also easier to model various parts of a system of interest modularly instead of all at once. Such an approach is conducive to the construction of a probabilistic model that tightly corresponds to a scientific system of interest, which is naturally thought of in separate components or modules. In a BHM, the rules of conditional probability can be used to specify the relevant distributions. For example, let us consider a mock system that has parameter vector $\theta$, an intermediate unobserved vector $S$, and observations $Y$. $\theta$ might be statistical or physical parameters, $S$ could be a quantity of scientific interest, and $Y$ could be noisy observations of $S$. A schematic for such a model is given in Figure 1, and the joint probability distribution is

$$p(Y, S, \theta) = p(\theta)p(S|\theta)p(Y|S, \theta).$$

The distribution $p(\theta)$ represents prior beliefs about parameters before data is collected, while $p(S|\theta)$ represents prior knowledge or assumptions for how $S$ is generated given parameters. For instance, this prior knowledge could entail clustering or some dependence between the elements of $S$. The process that models $Y$ conditional on $S$ and $\theta$ is $p(Y|S, \theta)$. The posterior distribution of scientific quantities of interest, $p(\theta, S|Y)$, is proportional to $p(Y, S, \theta)$ by Bayes' theorem. Estimates and assessments of uncertainty of scientific parameters and quantities can be extracted from the posterior distribution.

## 1.2 An Overview of Physical-Statistical Modeling with BHMs

The case for applying Bayesian hierarchical modeling and methodology in geophysics is strongly made by Berliner (2003), which he describes as "physical-statistical modeling". Particularly, employing the Bayesian hierarchical approach has the primary advantage of incorporating all relevant sources of uncertainty and randomness into one coherent probabilistic framework. The sources typically modeled together are: 1) measurement errors in the data collection process, 2) lack of full knowledge of the precise functional form of the underlying physical equations describing the physical phenomenon being modeled, or else

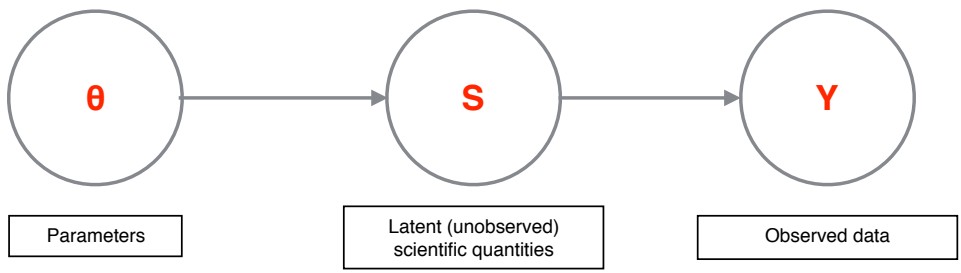

**Figure 1.** Schematic of a simple Bayesian hierarchical model; **here,** $\theta$ **represents physical parameters,** $S$ **represents unobserved scientific quantities of interest, and.** $Y$ **represents the observed data.**

simplification of the physical system description 3) numerical errors induced while approximating the solution to a system of partial differential equation PDEs, and 4) lack of precise knowledge of fundamental parameters (constants) in the underlying PDEs describing said phenomenon. In the Bayesian hierarchical framework **(Berliner, 1996; Wikle, 2016; Cressie and Wikle, 2015)** each of these sources of uncertainty is modeled by conditioning on the appropriate quantities, and inference is performed by sampling from or approximating the posterior distribution (the distribution of the unknown quantities of interest conditional on the observed data).

At the highest level of a BHM, prior probability distributions are laid out for the physical parameters of interest. At the intermediary level, a probability distribution for the physical process of interest is laid out conditional on the parameters, which is typically motivated by a numerical scheme for solving PDEs. In particular, this level may be modeled as the sum of the output from a numerical solver and an error correcting process. Finally, at the observed level, a probability distribution is set forth for the observed data conditional on the latent physical process and other relevant measurement parameters, which include variances of measuring procedures or devices. The product of these probability distributions specifies the joint distribution of all relevant quantities, which is proportional to the posterior distribution by the definition of conditional probability. While a traditional analysis may handle each of these disparate sources of uncertainty in an ad-hoc and disjointed fashion, the Bayesian hierarchical approach leverages probability measures to cohesively model major sources of uncertainty and undertake inference in a principled manner. Figure 2 diagrams what a prototypical physical-statistical Bayesian hierarchical model might look like.

**While the BHM approach to physical-statistical problems offers many advantages, it is not an infallible approach. In particular, while constructing a BHM may be straightforward, actually fitting a BHM to data can be computationally difficult. In the analysis that follows, there are only one to two physical parameters and the likelihood function is tractable, so posterior computation is not difficult. In more complex scenarios with many physical parameters (e.g., a basal sliding field with a parameter for each grid point), it becomes more difficult to compute the posterior or draw samples from it. There are now many new tools, however, for Bayesian inference of complicated and high dimensional posterior distributions, such as Stan (Stan Development Team, 2018) and INLA (Rue et al., 2017). Another potential difficulty in using BHMs for physical-statistical problems is that solving for a set of dynamical equations with a numerical**

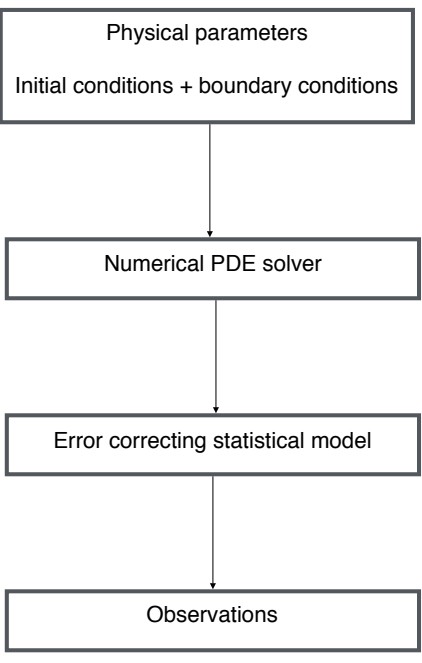

**Figure 2.** Schematic of a prototypical physical-statistical Bayesian hierarchical model. **At the top layer, physical parameters, initial conditions, and boundary conditions are fed into a numerical solver, and the output of this is corrected with an error correcting process; finally the actual observations are dependent on the actual physical process values.**

**method can be computationally onerous, generally speaking; while this is not a detriment in the work that follows, this can be a problem for posterior computation. One way to circumvent this issue is to emulate a numerical solver, using techniques as in Hooten et al. (2011). Another methodology that can be used to efficiently solve PDEs using Bayesian numerical analysis comes from Owhadi and Scovel (2017). Finally, Calderhead et al. (2008) suggests methodology to**
5  **avoid explicitly solving ordinary differential equations by using Gaussian processes.**

To put the contributions of this work into context, we briefly review glaciology papers that have employed Bayesian modeling. In Berliner et al. (2008), a Bayesian hierarchical approach is used to model ice streams in one spatial dimension, and **an error correcting process is utilized to account for a simplification in the physical model**. A combination of Markov chain Monte Carlo (MCMC) and empirical Bayes methodology is used to fit the model, and basal shear stress and resistive stresses

10  are included. Furthermore, wavelets are used for dimensionality reduction purposes so as to make the computations more feasible. In Pralong and Gudmundsson (2011), a Bayesian model is constructed for an ice stream where the likelihood and prior are Gaussian. The observed data are surface topography, horizontal and vertical surface velocities, and the latent system state is basal topography and slipperiness. The goal is to infer the system state given the observed data, and ultimately a maximum a posteriori (MAP) point estimate is used for inference in conjunction with an iterative method for posterior maximization.

Physics is incorporated by solving for the steady state solution with a finite element method (FEM) solver, given the system state. In Brinkerhoff et al. (2016) a flowline model of the SIA is considered with vertically integrated velocities. Gaussian process priors are used for all unknowns, and the Metropolis–Hastings algorithm is used to fit the model. The approach yields convincing results in simulations and a real data set. In Isaac et al. (2015), numerical methods are presented for solving a nonlinear Stokes equation boundary value problem for an ice sheet in Antarctica. The method ultimately uses a low rank approximation to a covariance matrix for the posterior distribution of a basal parameter field. Finally, and perhaps most directly related to this research, in Minchew et al. (2015) interferometric synthetic aperture radar (InSAR) is used to determine velocity fields at Langjökull and Hofsjökull in early June 2012. The velocity directions match the surface gradient, but magnitudes do not appear to coincide with the theoretical predictions of other authors (likely due to the inappropriate modeling of basal sliding).

**The same approach within this work can be used for non-SIA problems in cryosphere science, and the Bayesian hierarchical model does not necessitate analytical solutions; the analytical solutions are used for the evaluation of the particular BHM in the paper based upon the SIA. However, in general, the biggest difficulty will be in developing a statistical error correcting process that appropriately models numerical errors for an arbitrary scenario, where a numerical solver for a different set of dynamical equations is used. In the SIA context, we can rely on prior studies of Bueler et al. (2005) to tell us something about how the numerical errors will look like in the SIA case – i.e., spatial variation in the scale of numerical errors between the dome, interior, and margin. This error pattern will not hold in general for other geometries and systems, and so either different prior studies must be utilized, or if these don't exist, the hierarchical model must be extended to include a more general model for the error correcting process (e.g., a spatially varying field for the log of the scale of numerical errors with a Gaussian process prior).**

The main differentiating contribution of this paper is to utilize the exact analytical solutions from Bueler et al. (2005) to evaluate the BHM employed. An additional novelty is the derivation and utilization of a novel finite difference method for solving the SIA PDE that operates in two spatial dimensions; consequently, the Bayesian model employed also operates in two spatial dimensions, in addition to time. Finally, we explicitly model the errors due to a numerical solver with a spatio-temporal statistical process, which accounts for different scales of spatial variability within the dome, within the interior, and within the margin of the glacier, as well as accumulation of numerical errors forward in time.

## 2 Description of Models

### 2.1 Shallow ice approximation

The physics of glaciers is an extensive topic; hence, only the portions which are most relevant to this paper are described. The reader is pointed to the comprehensive works by Cuffey and Paterson (2010) and van der Veen (2017) for further reading on the subject. PDEs for glaciers are derived from the following considerations. First, glaciers are modeled as very slowly moving and viscous fluids. By applying the principle of mass conservation, the net ice flux moving in or out of an infinitesimal column of the glacier located at some spatial coordinate, plus the net mass change due to precipitation or melting, yields the change in

the height of the column over an infinitesimal time interval. Such a heuristic argument provides a PDE in two dimensions for a glacier, with averaged velocities in two spatial dimensions. The PDE relates the time derivative of the thickness of the glacier to the flux and net mass change (i.e., mass balance). The main assumptions are that ice is isotropic and homogeneous, and also that longitudinal and transverse stress terms can be ignored, which is reasonable when the overall thickness of the glacier is small in comparison to its width. Under these assumptions, the velocity of the ice is made up of two additive components. The first component of the velocity is based upon deformation due to gravity, which acts in the direction of steepest descent of the surface and is a function of the ice viscosity parameter. The second component of velocity also acts along the gradient of the glacier surface and is a function of the basal sliding parameter field. The formulations stem from Glen's flow law (Glen, 1955, 1958) and Weertman's sliding relation (Weertman, 1964).

Written in terms of glacial thickness, $H(x,y,t)$, the SIA PDE is:

$$
\begin{aligned}
H_t &= -[\bar{u}H]_x - [\bar{v}H]_y + \dot{b}. \\
-[\bar{u}H]_x &= -[-C_0\gamma(-\rho g H[H+R]_x)H + \frac{2B}{n+2}(\rho g \alpha)^{n-1}H^{n+1}(-\rho g H[H+R]_x)]_x \\
-[\bar{v}H]_y &= -[-C_0\gamma(-\rho g H[H+R]_y)H + \frac{2B}{n+2}(\rho g \alpha)^{n-1}H^{n+1}(-\rho g H[H+R]_y)]_y \\
\alpha &= \sqrt{[H+R]_x^2 + [H+R]_y^2}
\end{aligned}
$$

Here $H(x,y,t)$ is the thickness of the glacier at spatial coordinate $(x,y)$ and time $t$, $\bar{u}$ is the average velocity in the $x$ direction and $\bar{v}$ is the average velocity in the $y$ direction. This model is vertically integrated, and hence only two spatial dimensions are modeled. $R(x,y,t)$ is the bedrock elevation which is assumed to be constant in time, so it can be written as $R(x,y)$; $\dot{b}(x,y,t)$ is the mass balance field, $B$ and $C_0\gamma$ are physical parameters governing the viscosity and basal sliding; $\rho$ governs the mass density of the ice; and finally $n$ is Glen's flow law constant, typically set to 3. Initial conditions (i.e., $H(x,y,0)$) are assumed to be given, and the boundary condition $H \geq 0$ is assumed, just as in Table 2 of Bueler et al. (2005). Additional derivations and details on the SIA are covered in a variety of sources, including Fowler and Larson (1978), Hutter (1982), Hutter (1983), and Flowers et al. (2005).

It is important to make explicit that there are some limitations of this PDE. Besides ignoring longitudinal and transverse stress terms, the PDE does not model subglacial hydrology, tunneling systems, jökulhlaups, or surges, the dynamics of which are believed to contribute to dynamics of glaciers as a whole. Nonetheless, one hopes these equations may serve as a first approximation for shallow glacier dynamics. In addition to dynamics, another important physical consideration of glaciers is the relationship between temperature and viscosity, which follows an Arrhenius relationship (Cuffey and Paterson, 2010). However, in the context of Icelandic glaciers like Langjökull, this is not consequential since they are temperate (i.e., their temperature is at melting point).

## 2.2 Bayesian hierarchical model

In this section, we provide an overview and set–up of the BHM employed in addition to notation for the key parameters, both statistical and physical. The reader is referred, however, to Table 1 for a summary of the model parameters utilized and a

schematic illustrating the BHM in Figure 3. We use index $i$ to refer to spatial coordinates (for this model space is assumed to be discretized into squares) and index $j$ to refer to time coordinates. Furthermore, the notation $S_{\cdot,j}$ refers to the surface elevation at all spatial coordinates for a particular time index $j$. Keeping in line with the Bayesian hierarchical modeling framework from Wikle (2016) and Cressie and Wikle (2015), we delineate the models used for the data level, process level, and parameter

level. The primary inferential goals are to infer physical process parameters (i.e., ice viscosity and basal sliding) and to predict the height of the glacier at various time points and spatial locations besides those that have been observed (aligned to a grid for which we have bedrock and initial surface height conditions). Within the Bayesian framework, all inferential goals may be achieved by determining the posterior distribution of these quantities (i.e., their probability distributions conditioned on observed data).

At the *data level*, the observed height for each grid point is modeled with a normal distribution (denoted with the notation $\mathrm{N}(\mu,\tau^2)$, where $\mu$ is the mean and $\tau^2$ is the variance), where the mean is the physical process value, and the variance is assumed to be known. **In particular it is assumed that** $Y_{ij} \sim \mathrm{N}(S_{ij},\sigma^2)$, where $Y_{ij}$ is the observed surface elevation of the glacier at location $i$ and time index $j$, $S_{ij}$ is the latent (i.e., unobserved) surface elevation at location $i$ and time index $j$ (equivalent to sum of the glacier thickness and bedrock level), and $\sigma^2$ is the variance of the measurement errors for the surface height

observations, **a fixed a and known quantity**. The number of observed spatial indices is assumed to be much smaller than the number of total spatial indices modeled at the latent level.

At the *process level*, $S_{\cdot,j} = f(S_0, B, \dot{b}, C_0\gamma, j) + X_j$, where $f$ is a numerical solution to the SIA at time index $j$, and $X_j$ is an error-correcting process at time index $j$. A finite difference version of the SIA PDE is described in full detail in Appendix A. In principle, however, the function $f$ may be derived from other numerical solvers. Additionally, it should be made clear that $f$ is

the output of a numerical solver for the underlying dynamics Also, $S_0$ denotes the glacier surface elevation values at the initial time point, which are assumed to be known; e.g., with high precision light detection and ranging (LIDAR) initial conditions provided by the Institute of Earth Sciences at the University of Iceland. $\dot{b}_{\cdot,j}$ is the mass balance field for time index $j$ at all the grid points, which is assumed to be fixed and known for the purpose of this analysis. $B$ is the ice viscosity parameter and $C_0\gamma$ is the basal sliding field, which itself is parametrized with $\mu_{\max}$ as in equation (16) of Bueler et al. (2005) and, furthermore, is

static in time. For compact notation, $\theta$ is used to refer to $B$ in test cases B-D and $(B, \mu_{\max})$ jointly in test case E.

Since we believe numerical errors will accumulate over time (Bueler et al., 2005), we define the error correcting process as follows: $X_{j+1} = X_j + \epsilon_{j+1}$, where $\epsilon_{j+1}$ is $MVN(0, \Sigma)$. (MVN stands for multivariate normal, and the first argument is the mean and the second is the covariance.) $\Sigma$ is block diagonal, with three blocks for indices corresponding to the margin, interior, and dome of the glacier (the margin is defined as the last grid squares before the glacier drops to 0 thickness, and

the dome is the origin grid square), respectively. Each block is defined from a square-exponential kernel with the same length scale, denoted by $\phi$, but distinct marginal variances, $\sigma_{\text{interior}}^2$, $\sigma_{\text{margin}}^2$ and $\sigma_{\text{dome}}^2$. The motivation for using different marginal variance parameters is to account for the widely different errors exhibited at the dome, interior, and margin, as is demonstrated by Bueler et al. (2005) and Jarosch et al. (2013). This error correcting process leads to a tractable likelihood function, as is shown in Appendix B.

Finally, at the *parameter level*, $B$ and $\mu_{\max}$ are endowed with truncated normal distributions as priors. $B$ has a normal prior with mean $3.5 \times 10^{-24}$, standard deviation $3 \times 10^{-24}$, truncated to have support $[1, 70] \times 10^{-24}$. $\mu_{\max}$ has a normal prior with mean $3 \times 10^{-11}$ and standard deviation $1 \times 10^{-11}$, truncated to have support $[1, 70] \times 10^{-12}$. (Units are $s^{-1}Pa^{-3}$ for ice viscosity and $Pa^{-1}ms^{-1}$ for basal sliding.) **The prior supports for $B$ and $\mu_{max}$ provide** plausible values for temperate ice

caps.

It is prudent to discuss the motivations and justifications of the various modeling choices employed in the model previously delineated. The data level is assumed to have independent normal errors with fixed variance; this is justified because of the uniformity of the measuring technology used from site to site (e.g., digital GPS) and symmetry of errors. On the other hand, the precise functional form of the data level is chosen somewhat arbitrarily as a Gaussian, which affords one analytical convenience.

Similarly, the error correcting process at the process level uses a zero mean Gaussian process with a parameterized covariance kernel (e.g., square exponential), mostly as an analytically manageable way to induce spatial correlation in the error correcting process. Spatial correlation in numerical errors has been demonstrated, for example, in Bueler et al. (2005).

Moreover, it is appropriate to consider potential variations of this model for slightly different scenarios; naturally, these could fall into: alternate choices of covariance kernel at the process level (e.g., Matérn, to allow for a less smooth error correcting

process) and varying errors at the data level, for example to account for compaction or densification that occurs between seasons. For the latter, a suggestion is to use conjugate inverse-gamma distributions for the variances, so that sampling can be accomplished with a Gibbs sampler. Additionally, as aforementioned, one can conceivably use any numerical solver for a PDE at the process level. Future variations may consider utilizing non-zero mean Gaussian processes for the error correction process, which may be more computationally costly yet perhaps more realistic. Generally, this model can be adapted to any science

or engineering system that is driven by physically meaningful parameters, whose dynamics are solved by noisy numerical methods, and for which noisy and continuous data is collected with well probed errors.

The mathematical details for how to do posterior computation within this model are given in Appendix B, which includes a derivation of an approximation to the log-likelihood that allows for computational efficiency. In summary, we compute the posterior of physical parameters directly on a grid since there are at most two physical parameters, and we use samples from

the posterior distribution of physical parameters to generate predictions for glacier thickness in the future.

## 3   Experiments to assess the Bayesian hierarchical model

### 3.1   Analytical solutions

In Bueler et al. (2005), analytical solutions to the SIA are presented as benchmarks for numerical solvers of the SIA. As opposed

to using other benchmarks such as the EISMINT experiment (Payne et al., 2000), which itself is based on numerical modeling and hence subject to numerical errors, the benchmark solutions provided in this work can be treated as ground truth to compare to. (This is in the sense that these are exact solutions of the SIA, but it must be stressed that the SIA is an approximation of the true physical dynamics governing a glacier.) These analytical solutions serve as a basis for simulating data sets to validate

| Parameter Name | Symbol | Description |
|---|---|---|
| Time index | $j$ | A subscript which refers to discrete time points |
| Spatial index | $i$ | A subscript which refers to discrete spatial points |
| All spatial points for a time index | $.,j$ | Refers to entire spatial field at time $j$ |
| ice viscosity | $B$ | Key physical parameter driving the SIA |
| Basal sliding | $C_0\gamma$ | Basal sliding field and key parameter driving the SIA |
| Max basal sliding | $\mu_{\mathrm{max}}$ | Parameter for the basal sliding field of test case E in Bueler et al. (2005) |
| Physical parameters | $\theta$ | Refers to physical parameters |
| Measurement error | $\sigma$ | Measurement error of surface elevation measurements |
| Error correcting covariance matrix | $\Sigma$ | Covariance matrix used for the error correcting process |
| Error correcting parameters | $(\sigma_{\mathrm{dome}}, \sigma_{\mathrm{interior}}, \sigma_{\mathrm{margin}}, \phi)$ | Parameters corresponding to $\Sigma$ |
| Mass balance field | $\dot{b}_{.,j}$ | Mass balance field at time index $j$ |
| Initial surface elevation | $S_0$ | Initial surface height of the glacier |

**Table 1.** A summary of main parameters and notation utilized.

the Bayesian hierarchical approaches developed in this paper. In other words, the exact analytical solutions provide the latent process in the BHM, conditioning on given initial conditions and mass balance functions. Hence to simulate data from the BHM, one can bypass the need to numerically solve the PDE and introduce errors.

We make use of four analytical solutions from Bueler et al. (2005) that are summarized here, but the reader is referred to the
original paper for the exact mathematical formulation and derivation of these analytical solutions. All of the analytical solutions assume a flat bedrock. Test case B includes no mass balance or basal sliding, and, consequently, the motion of the glacier is only attributable to deformation due to gravity. Test case C makes use of a mass balance field that is inversely proportional to time and directly proportional to thickness, but there is no basal sliding field modeled. Similarly, test case D utilizes a mass balance field with no basal sliding field modeled. In distinction from test case C, however, the mass balance field of test case
D is such that the overall solution for glacial thickness is periodic in time. Finally, in contrast to the other tests, test case E has a spatially varying basal sliding field, yet the overall solution is static in time. Note that test A was not utilized in this study because it is a steady state solution without a varying mass balance or basal sliding field.

### 3.2 Simulation study test details

Conditions of the simulation study have been chosen as to closely emulate the data collected at Langjökull ice cap by the
Institute of Earth Sciences at the University of Iceland (IES-UI). In particular, 20 years of data are assumed, which is comparable to data provided by the IES. 25 fixed measurement sites are used for bi-annual surface elevation measurements, which are geographically distributed on the glacier in a manner that is comparable to the real data provided by the IES-UI. **Figure 4 illustrates the locations of these measurement sites on the glacier.** Surface elevation measurements for these sites are taken

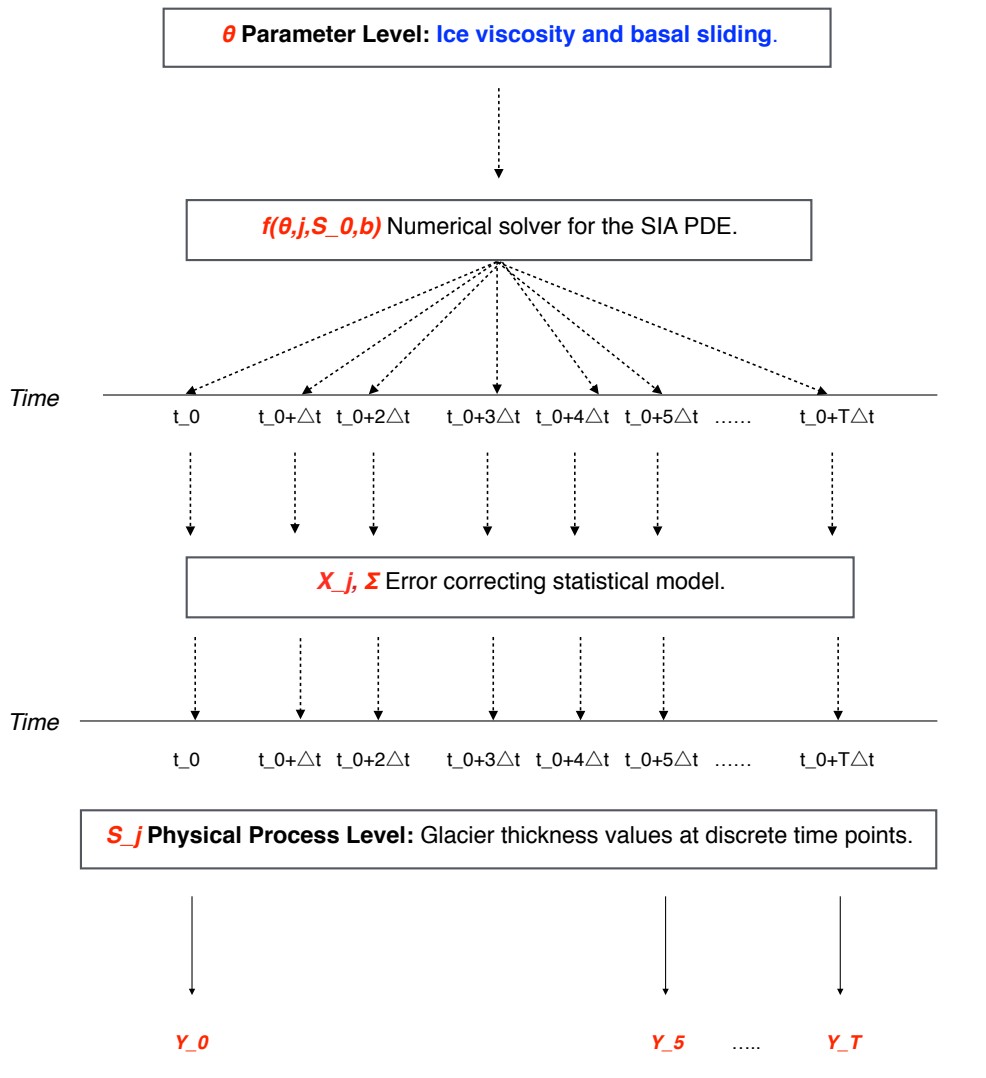

**Figure 3.** Schematic of the physical-statistical BHM that has been constructed based on the SIA PDE. The main parameters and variables for each module of the physical-statistical model are highlighted in red. **The main levels of a physical-statistical model shown in Figure 2 are displayed here, along with the parameters and variables describing each level.**

twice a year (i.e., for summer and winter mass balance measurements). The surface elevation measurements are generated by adding Gaussian noise (zero mean, unit variance) to the analytical solutions at the spatio-temporal coordinates of the fixed measurement sites. The choice of unit variance is larger than the errors produced by digital-GPS measurements. Remaining

physical parameters were chosen using the values from Bueler et al. (2005) Table 2 to allow for comparisons to this work and the EISMINT I experiment (Payne et al., 2000).

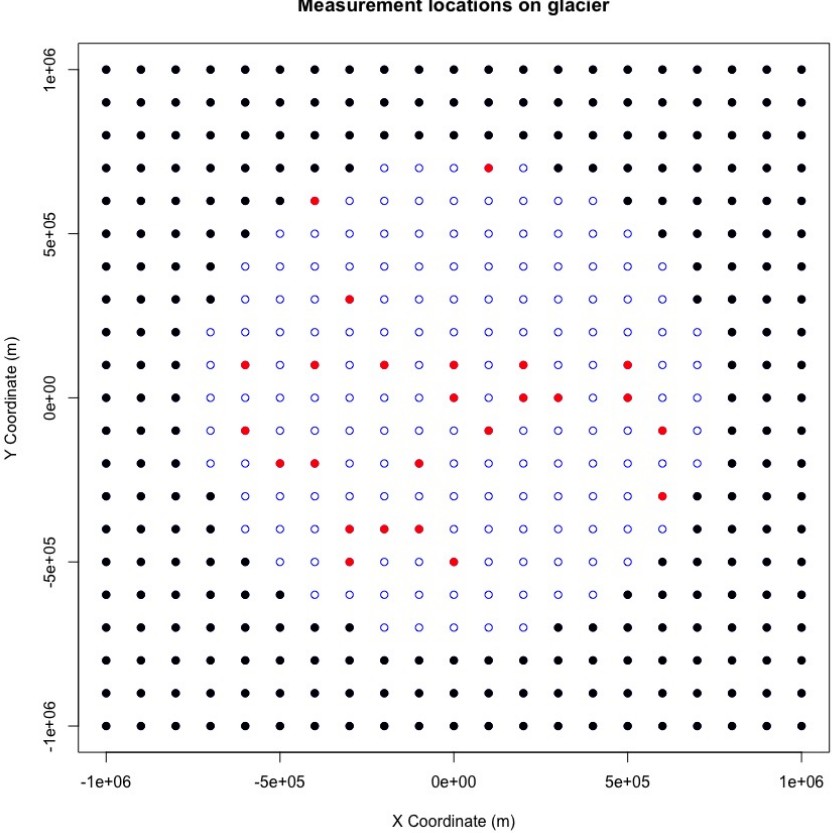

**Figure 4.** An illustration marking the 25 measurement sites on the glacier. This is a top level view of the glacier, where the blue points indicate the glacier, the red points indicate the measurement locations, and the black points indicate locations surrounding the glacier with no glacial thickness.

## 4 Results

Validation and diagnostics of the BHM were achieved through a combination of an assessment of posterior probability intervals, a test of the predictive error of thickness values 100 years from the initial time point $t_0$, and a comparison between observed and expected values for numerical errors based on the error correcting process utilized. As is discussed in more detail below, these assessments suggest that the BHM is useful for inference of posterior probability distributions for physical parameters,

prediction of future glacial thickness values on the order of 100 years, and the modeling of numerical errors at the margin, interior, and dome of the glacier.

Table 2 contains posterior credibility intervals for ice viscosity in test cases B-D. **A 3-sd credibility interval was computed with mean +/- 3 standard deviations of the posterior samples. In all of these test cases, the 3-sd credibility interval covers the actual ice viscosity.** Furthermore, as is apparent in Table 3, the predictive error, relative to thickness values on the order of a kilometer, appears be small overall, particularly at the interior; predictive error is the root mean squared difference between predictions and the exact analytical values for each of the test cases. Note that test E was not included with the predictive checks since it is static in time. Consistent with Bueler et al. (2005) and Jarosch et al. (2013), however, errors are greatest at the margin and dome of the glacier (evident in Figure 6). Nonetheless, the predictive distributions cover the actual thicknesses even at these extremes. This illustrates the utility of the BHM for accounting for errors induced by the numerical solution of the SIA. Additionally, an illustration comparing the posterior and prior distributions for test case D is shown in Figure 7.

To investigate the frequentist properties of the posterior probability distribution for ice viscosity (i.e., its performance under repeated sampling of data), 500 simulations were completed under repeated sampling of the surface elevation data at the 25 fixed measurement sites for test cases B-D. The coverage of ice viscosity for a 3-sd interval was computed for each of the simulations, where coverage for a given interval is binary; either the actual parameter value is in the interval or it is not. **For test case B, in 499 of 500 simulations the 3-sd credibility interval covered the actual value of ice viscosity. In test cases C and D, the 3-sd credibility interval covered the actual value of ice viscosity in all of the simulations. This suggests that the frequentist coverage probability of the credibility interval is at least 99 percent.**

For test case E, one assumes that the field is described by parameterized equation (16) of Bueler et al. (2005). That is, in polar coordinates with radius $r$ and angle $\Theta$:

$$C_0\gamma(r,\Theta) = \frac{\mu_{\max}4(r-r_1)(r_2-r)4(\Theta-\theta_1)(\theta_2-\Theta)}{(r_2-r_1)^2(\theta_2-\theta_1)^2}$$

for $\theta_1 < \Theta < \theta_2$ and $r_1 < r < r_2$, and $C_0\gamma = 0$ otherwise. In addition to ice viscosity, the inferential object of interest is the scale parameter $\mu_{\max}$. **The 3-sd posterior credibility interval for $B$ is $[1,43]$ in units of $10^{-25}s^{-1}Pa^{-3}$, and for $\mu_{\max}$ it is $[1,50]$ in units of** $10^{-12}Pa^{-1}ms^{-1}$. The actual values for $B$ and $\mu_{\max}$ are $32\times10^{-25}s^{-1}Pa^{-3}$ and $25\times10^{-12}s^{-1}Pa^{-1}ms^{-1}$, respectively. Hence, the credibility intervals cover both parameters. A figure illustrating the posterior distribution of $\mu_{max}$ is given in the supplemental materials.

**While the credibility intervals achieved coverage of the actual values of the parameters, it was noticed that the posterior distribution for physical parameters and predictions are biased. Brynjarsdóttir and O'Hagan (2014) exhibit the same phenomenon in a simple physical system with a single physical parameter, and they demonstrate that the bias of a physical parameter posterior distribution reduces as better prior information is encoded to model the difference between the output of a computer simulator of a physical system and the actual physical process values (i.e., what we have termed as an error correcting process). To demonstrate that this also holds in the BHM presented in this paper, we consider the following comparison. To assign prior information to the error correcting process, we consider a discrete parameter set for $\sigma^2_{\text{interior}}$, $\sigma^2_{\text{margin}}$ and $\sigma^2_{\text{dome}}$: {.1,1,10,100} in units of $m^2$, which corresponds to different orders of**

magnitude for variability. In one case, we ignore prior information from Bueler et al. (2005) and put equal probability mass on the parameter space for these parameters. In the second case, we encode more realistic prior information into the scales of errors at the three regions: equal mass on 10 and 100 at the margin, equal mass on .1 and 1 at the interior, and equal mass at 1 and 10 at the dome (all units are $m^2$). In both cases, the parameter $\phi$ is fixed at 70 km to place emphasis on the scales of error. The results of inferring the posterior distribution for ice viscosity $B$ are shown in Figure 8. Consistent with Brynjarsdóttir and O'Hagan (2014), the posterior distribution of the physical parameter $B$ is much less biased when prior information is encoded into the error correcting process.

To assess how the posterior distribution for ice viscosity evolves under different sampling plans of the data, we conducted a series of simulations in test case D under varying sampling periods. In particular, we considered data samples once every 10 years, once every 5 years, once a year, and twice a year; the resulting posteriors for ice viscosity are in Figure 9. The general pattern is that the bias of the posterior distributions reduces as the period gets shorter, although the posterior becomes more diffuse. The result that some posterior uncertainty does not go away with more collected data is also consistent with the results in Brynjarsdóttir and O'Hagan (2014). The particular period we chose in this analysis (data collected twice a year) was meant to model how the UI-IES Glaciology Team collects data, that is, twice a year due to summer and winter mass balance measurements.

To assess the accumulating error-correcting process model, we estimated the marginal variances of the error correcting process for each of the time points with observed data in test case B, by examining the residuals formed by the difference between the numerical solver and the observed data. According to the model, the standard deviation of these residuals at the interior of the glacier should grow as $\sqrt{\sigma^2 + t\sigma^2_{\text{interior}}}$, where $t$ is the number of time steps (and likewise at the dome and margin). Figure 10 shows a match between observed and expected in this regard, and, in particular, the 99 percent confidence bands appear to cover the expected variability as time progresses. Also apparent from this figure is that, as time progresses, the errors at the margin, dome, and interior contribute more error than measurement error, which is on the order of 1 meter. Moreover, this is also evident in Table 4, since after 200 time steps from $t_0$ (i.e., 20 years), the marginal variances will be $200\sigma^2_{\text{interior}}$, $200\sigma^2_{\text{margin}}$, and $200\sigma^2_{\text{dome}}$ based on the accumulating errors model; all of these values exceed 1, the measurement variance.

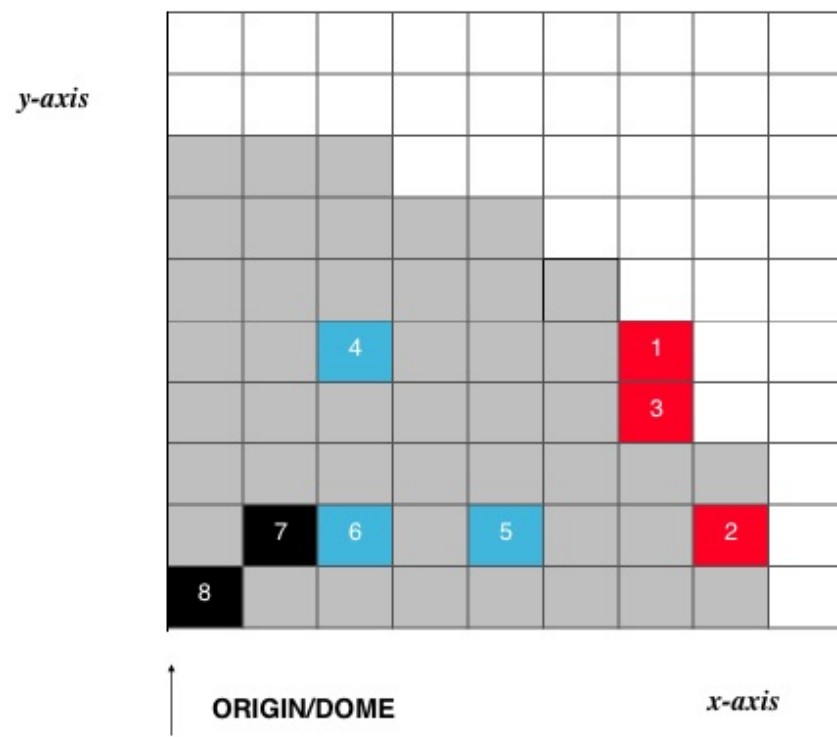

**Figure 5.** Grid map used to interpret the following box-plots in Figure 6. Eight randomly chosen grid points are selected **for testing predictions; these are not the same as the measurement locations**. Only one quadrant of the glacier is shown due to symmetry as is done in Figures 9,10, and 12 of Bueler et al. (2005), and the width of each cell is $10^5$ m. Additionally, the red squares indicate locations at or close to the margin, the blue squares indicate locations that are between the dome and margin of the glacier, and the black squares indicate locations at or close to the dome of the glacier. Moreover, glacier grid squares with non-zero thickness are shaded in grey, as to indicate the glacier location.

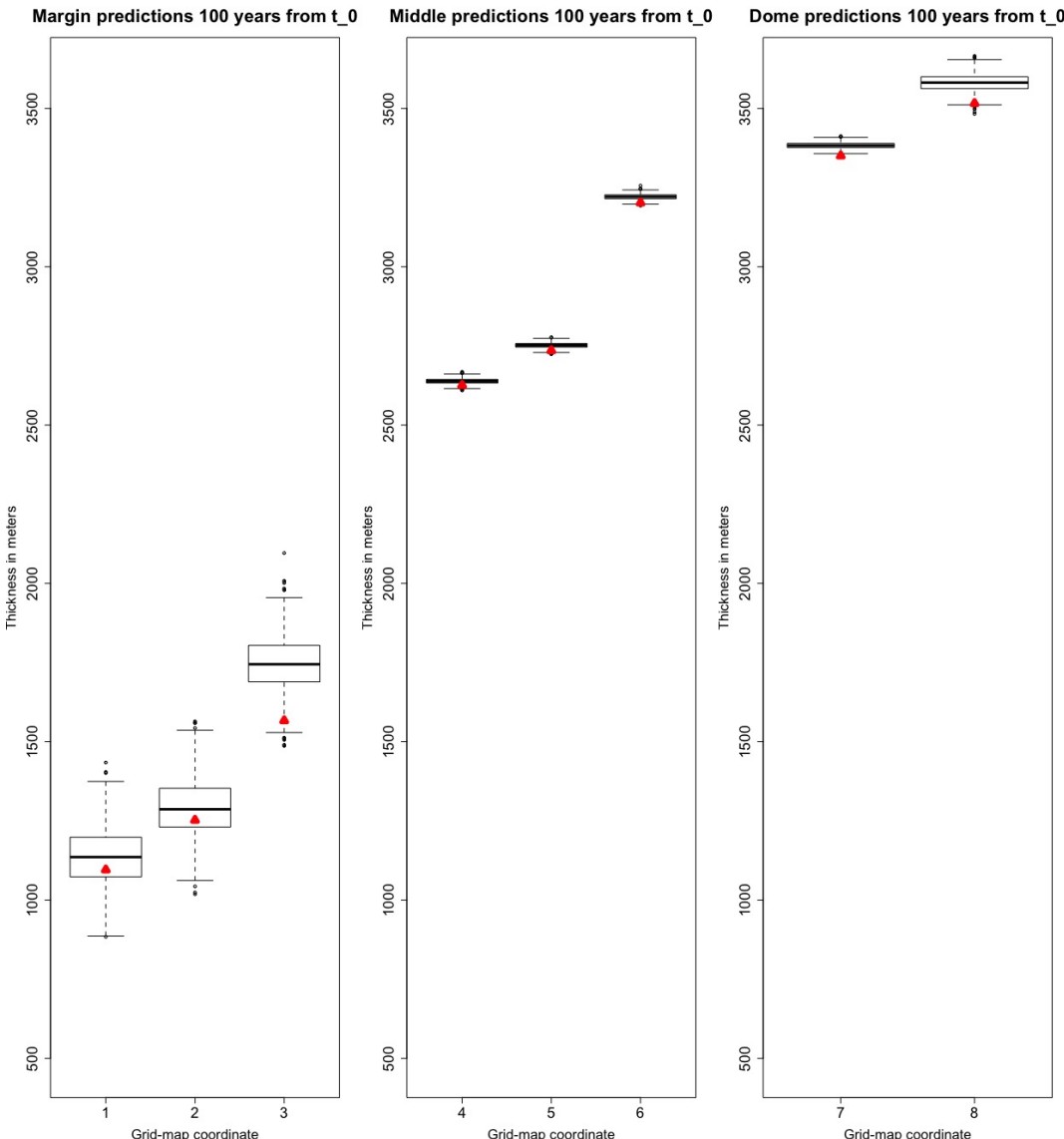

**Figure 6.** Thickness prediction samples 100 years from $t_0$ for test case B **(i.e., no mass balance field or basal sliding)**. Triangles indicate the actual thickness values from the analytical solution. The first set of plots are close to the margin (red squares of Figure 5), the second set of plots are between the dome and margin of the glacier (blue squares of Figure 5), and the final set of plots are towards the dome of the glacier (black squares of Figure 5). Refer to Figure 5 for a grid map to spatially reference each of the boxplots. As can be expected according to Bueler et al. (2005), largest errors occur at the dome and the margin. Note on interpretation: the middle of each box is the median, the interquartile range is denoted by the box, and **1.5 of the interquartile range beyond the first and third quartile is illustrated with the whiskers. Those points that are more than 1.5 of the interquartile range beyond the first and third quartiles are outliers, and they are denoted with circles.**

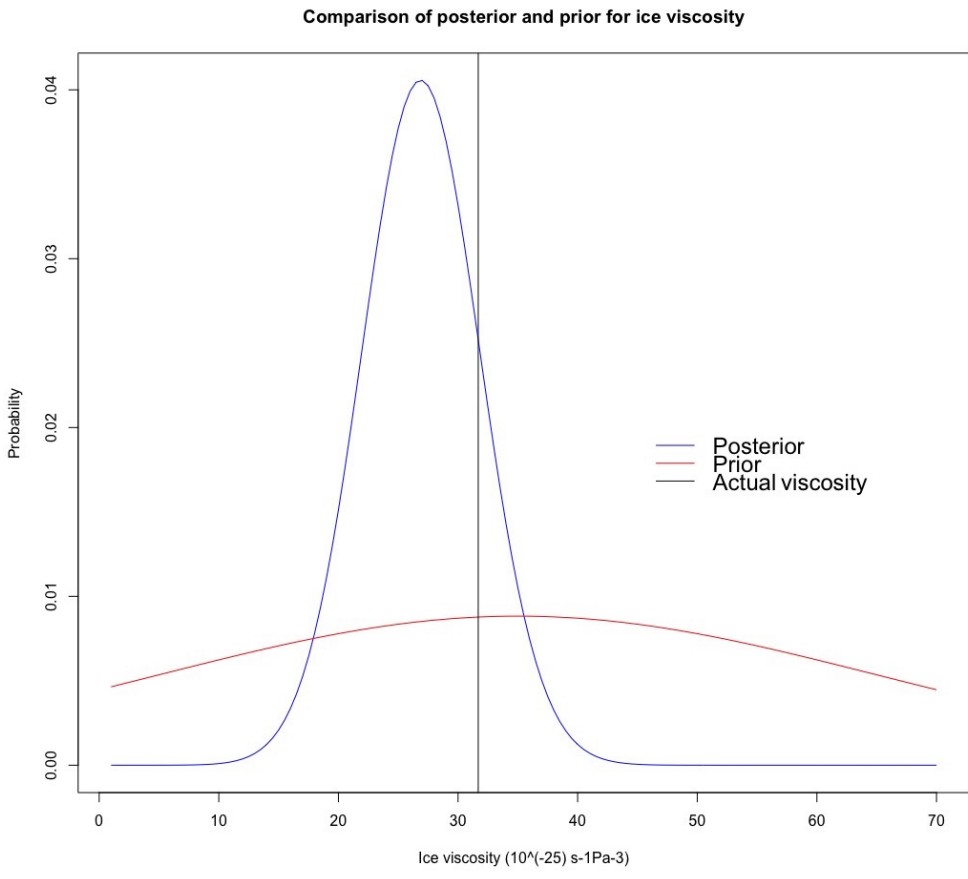

**Figure 7.** Comparison of posterior and prior distributions of ice viscosity for test case D (**i.e., mass balance field producing a periodic SIA solution**).

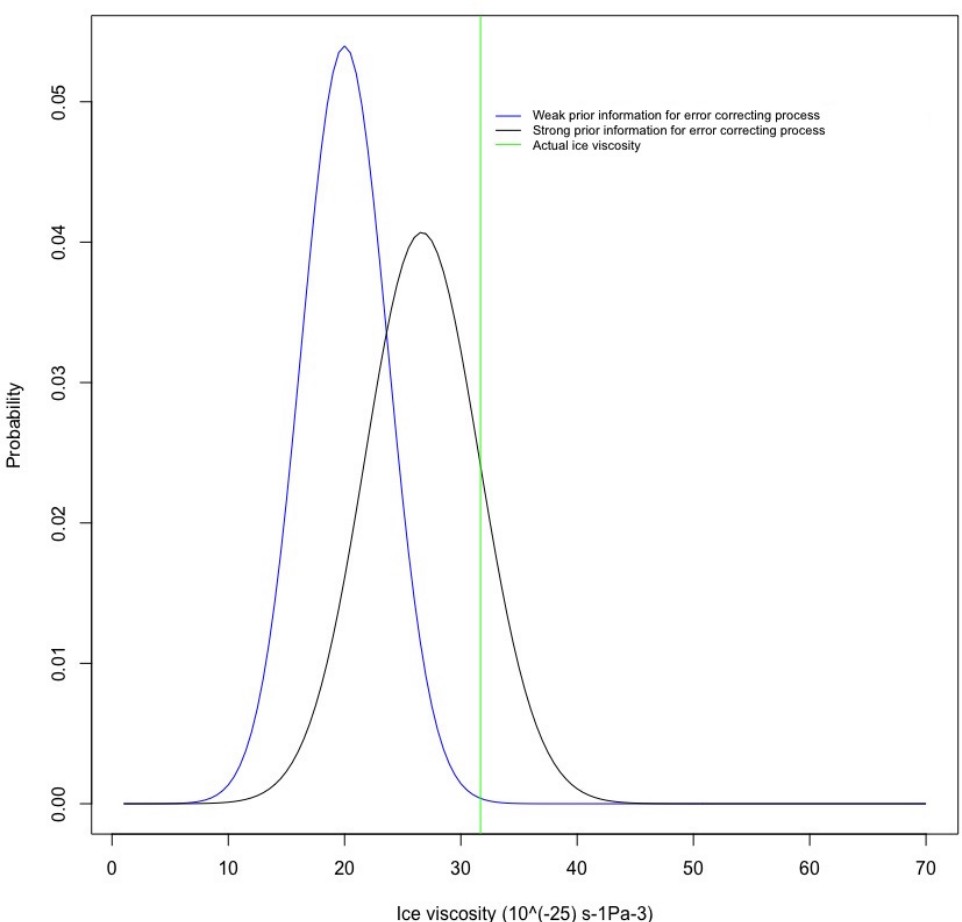

**Figure 8.** A comparison of posteriors under strong and weak prior information for the error correcting process in test case D (i.e., mass balance field producing a periodic SIA solution); prior information for the error correcting process leads to a less biased posterior, though with slightly more posterior uncertainty.

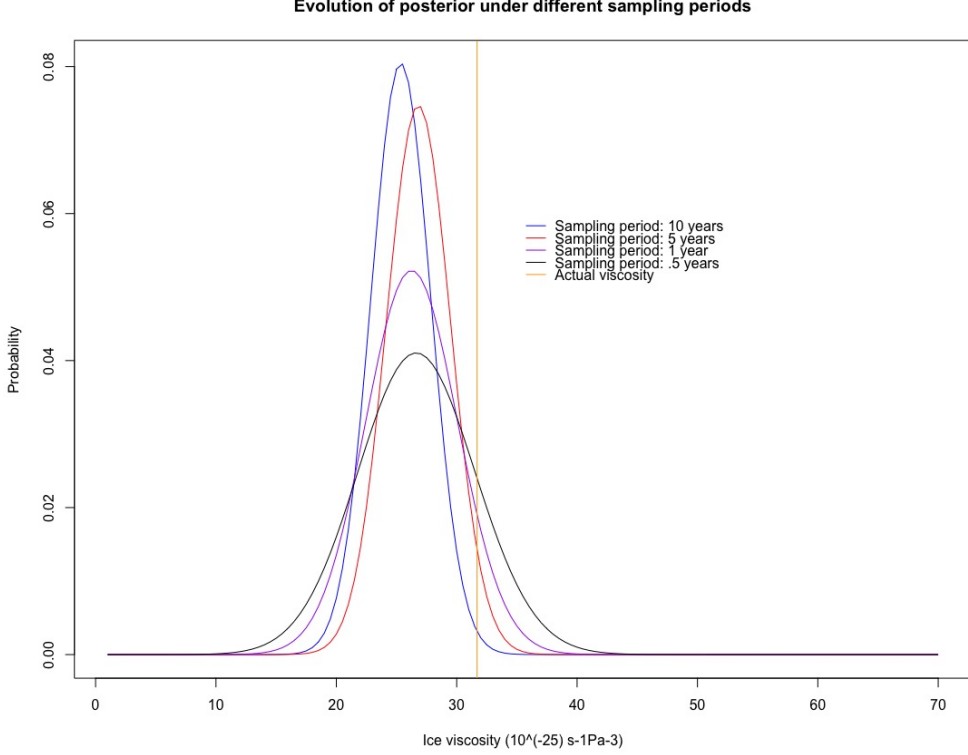

**Figure 9.** A comparison of posteriors in test case D (i.e., mass balance field producing a periodic SIA solution) under different sampling periods: data sampled once every 10 years, every 5 years, once a year, and twice a year. The general trend is that the posterior tends to become less biased as the period of sampling decreases, although the posterior becomes more diffuse. The University of Iceland Institute of Earth Sciences Glaciology Team takes measurements twice a year for summer and winter mass balance measurements.

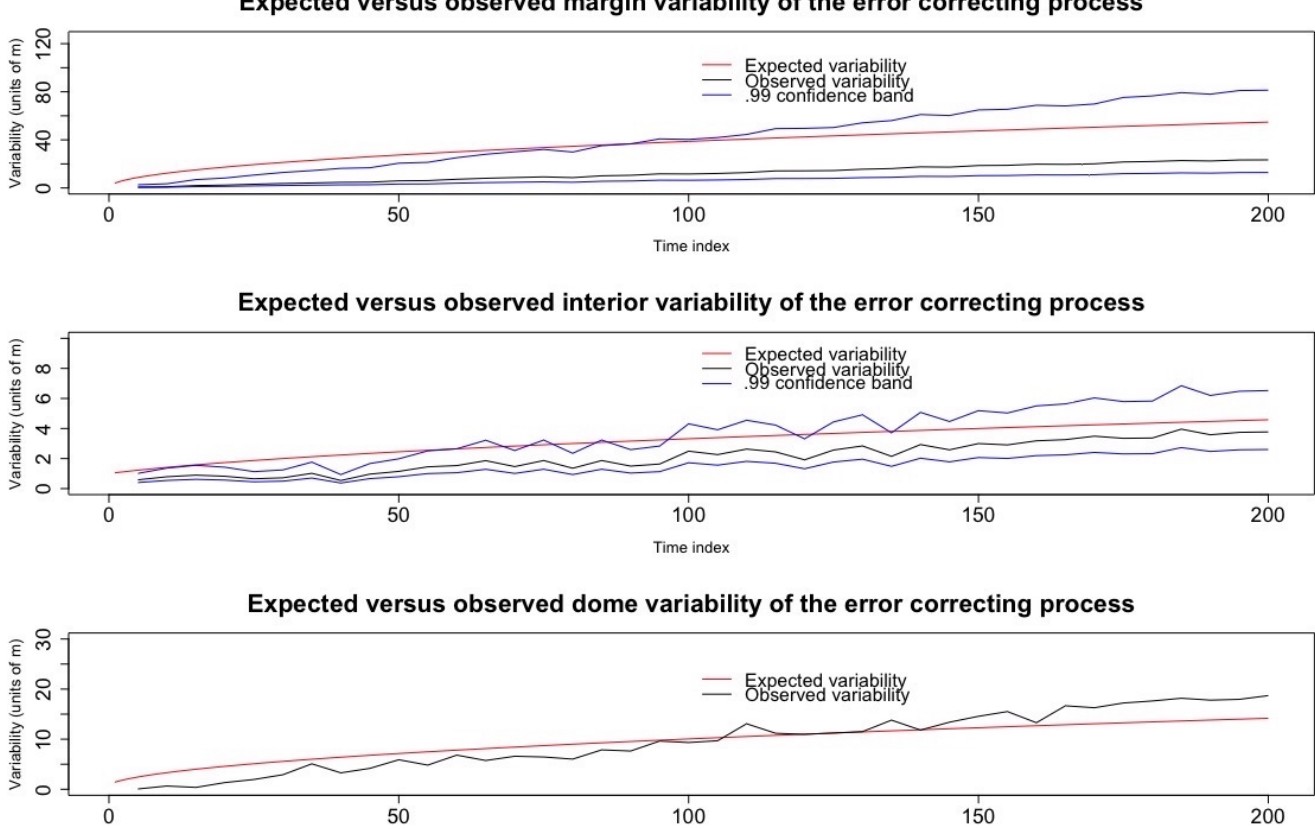

**Figure 10.** An illustration comparing the expected variability of the error correcting process (as per the Bayesian hierarchical model) to the observed variability of residuals at the interior, margin, and dome for **test case B (i.e., no mass balance field or basal sliding). These residuals are the differences between the observed data and the numerical solution.**

| Test Case | Actual Viscosity | 3-sd Credibility Interval |
|---|---|---|
| Bueler B | 32 | $[7, 34]$ |
| Bueler C | 32 | $[5, 33]$ |
| Bueler D | 32 | $[11, 42]$ |
| **Units** | $10^{-25}\ s^{-1} Pa^{-3}$ | $10^{-25}\ s^{-1} Pa^{-3}$ |

**Table 2.** Ice viscosity posterior intervals.

| Test Case | Dome RMSE | Interior RMSE | Margin RMSE |
|---|---|---|---|
| Bueler B | 66 | 20 | 75 |
| Bueler C | 76 | 22 | 82 |
| Bueler D | 1.4 | 17 | 49 |
| **Units** | m | m | m |

**Table 3.** Results of prediction at $t_0 + 100$ years. **RMSE stands for root mean squared error. This is calculated by taking the average of the squared difference between the actual glacial thickness values and predicted glacial thickness values, and then taking the square root.**

| Test Case | $\sigma^2_{\text{dome}}$ | $\sigma^2_{\text{interior}}$ | $\sigma^2_{\text{margin}}$ | $\phi$ |
|---|---|---|---|---|
| Bueler B | 1 | .1 | 15 | 71 |
| Bueler C | 1 | .15 | 15 | 64 |
| Bueler D | .1 | .1 | 10 | 62 |
| Bueler E | .1 | .1 | 10 | 60 |
| **Units** | sq. m | sq. m | sq. m | km |

**Table 4.** Error correcting process hyper-parameters; $\sigma^2_{\text{dome}}$ **is the error correcting process variance at the dome,** $\sigma^2_{\text{interior}}$ **is the error correcting process variance at the interior,** $\sigma^2_{\text{margin}}$ **is the error correcting process variance at the margin, and** $\phi$ **is the length scale parameter.**

## 5 Summary, discussion, and future work

The primary contribution of this work has been to construct a BHM for glacier flow based on the SIA that operates in two spatial dimensions and time, which successfully models numerical errors induced by a numerical solver that accumulate with time and vary spatially. This BHM leads to full posterior probability distributions for physical parameters as well as a principled method for making predictions that takes into account both numerical errors and uncertainty in key physical parameters. Furthermore, the BHM operates in two spatial dimensions and time, which, to our knowledge, is new to the field of glaciology. An additional contribution is the derivation of a novel finite difference method for solving the SIA. When tested using simulated data sets based on analytical solutions to the SIA from Bueler et al. (2005), the results herein indicate that our approach is able to infer meaningful probability distributions for glacial parameters, and, furthermore, this approach makes probabilistic predictions for glacial thickness that adequately account for the error induced by using a numerical solver of the SIA. A future goal is to create an R package for fitting a generalized version of the model used within this work, where the function $f(.)$ is provided by the user. This will allow glaciologists to extend the modeling approach we have developed to other similar scenarios in which the physical dynamics are more complex than the SIA. An additional scenario for which this package can be useful is when the numerical method is not a finite difference method; e.g., a FEM. To this end, we will attempt to utilize emulator inference (Hooten et al., 2011); this will be crucial to ensure that the methodology scales well computationally, since each posterior sample requires a forward PDE solve. Finally, and perhaps most importantly, future work will involve the application of the modeling and methodologies developed within this paper to real data collected by the IES-UI, which includes bedrock elevation and mass balance measurements.

*Author contributions.* All of the glaciologists contributed equally to this work.

*Acknowledgements.* The Icelandic Research Fund (RANNIS) is thanked for funding this research.

## Appendix A: Finite difference method for the shallow ice approximation

Here a finite difference scheme is derived for the SIA PDE. The overarching strategy in developing this finite discretization scheme is to take a second order Taylor expansion for $H(x, y, t)$ with $x, y$ fixed, and then equate the resultant time derivatives, $H_t$ and $H_{tt}$, to functions of spatial derivatives by using the original SIA PDE. That is, one starts with the approximation $H(x, y, t + \Delta t) \approx H(x, y, t) + H_t(x, y, t)\Delta t + H_{tt}(x, y, t)\Delta t^2/2$ and uses the first equation of section two to write $H_t$ and $H_{tt}$ in terms of spatial derivatives. Finally, central differences in space are substituted for the spatial derivatives. This finite difference scheme is motivated by the Lax-Wendroff (Hudson) method, which is generally better than finite difference methods that use only a single order Taylor expansion (indeed, in the advection-diffusion equation such methods may be unconditionally unstable).

In the following derivations note that the subscripts mean 'derivative with respect to' (e.g., $H_t$ means derivative of $H$ with respect to $t$).

$$H_t = -[\bar{u}H]_x - [\bar{v}H]_y + \dot{b}$$
$$H_{tt} = -[\bar{u}H]_{xt} - [\bar{v}H]_{yt} + \ddot{b}.$$

Now we solve for these derivatives in terms of spatial derivatives in $H(x,y,t)$, the glacier thickness, and $R(x,y)$, the bedrock level. The derivation makes repeated use of the differentiation rule for products, the chain rule for differentiation, and equality of mixed partials (e.g., $H_{xt} = H_{tx}$).

$$-[\bar{u}H]_x = -C_0\gamma\rho g T_1 + \frac{2B}{n+2}(\rho g)^n T_2$$
$$T_1 = [2HH_x(H_x + R_x) + H^2(H_{xx} + R_{xx})]$$
$$T_2 = [[\alpha^{n-1}]_x[H^{n+2}H_x + H^{n+2}R_x] + \alpha^{n-1}[(n+2)H^{n+1}H_x^2 + (n+2)H^{n+1}H_xR_x + H^{n+2}H_{xx} + H^{n+2}R_{xx}]]$$

By symmetry in $x$ and $y$, $-[\bar{v}H]_y$ can be analogously derived:

$$-[\bar{v}H]_y = -C_0\gamma\rho g T_3 + \frac{2B}{n+2}(\rho g)^n T_4$$
$$T_3 = [2HH_y(H_y + R_y) + H^2(H_{yy} + R_{yy})]$$
$$T_4 = [[\alpha^{n-1}]_y[H^{n+2}H_y + H^{n+2}R_y] + \alpha^{n-1}[(n+2)H^{n+1}H_y^2 + (n+2)H^{n+1}H_yR_y + H^{n+2}H_{yy} + H^{n+2}R_{yy}]]$$

Derivatives $[\alpha^{n-1}]_x$ and $[\alpha^{n-1}]_y$:

$$[\alpha^{n-1}]_x = \frac{n-1}{2}(S_x^2 + S_y^2)^{\frac{n-3}{2}}(2S_xS_{xx} + 2S_yS_{yx})$$

$$[\alpha^{n-1}]_y = \frac{n-1}{2}(S_x^2 + S_y^2)^{\frac{n-3}{2}}(2S_yS_{yy} + 2S_xS_{xy})$$

Now we derive $-[\bar{u}H]_{xt}$

$$
\begin{aligned}
-[\bar{u}H]_{xt} &= -C_0\gamma\rho g T_{1t} + \frac{2B}{n+2}(\rho g)^n T_{2t} \\
T_{1t} &= [2H_t H_x^2 + 4HH_x H_{xt} + 2HH_{xt}R_x + 2HH_x R_{xt} + 2H_t H_x R_x + 2HH_t H_{xx} + H^2 H_{xxt} + 2HH_t R_{xx} + H^2 R_{xxt}] \\
T_{2t} &= [T_5 + T_6 + T_7 + T_8] \\
T_5 &= [\alpha^{n-1}]_{xt} H^{n+2} H_x \\
T_6 &= [\alpha^{n-1}]_{xt} H^{n+2} R_x \\
T_7 &= [\alpha^{n-1}]_x[(n+2)H^{n+1}H_t H_x + H^{n+2}H_{xt} + (n+2)H^{n+1}H_t R_x + H^{n+2}R_{xt}] \\
T_8 &= [\alpha^{n-1}]_{xt} H^{n+2}H_x + \alpha_x^{n-1}(n+2)H^{n+1}H_t H_x + \alpha_x^{n-1}H^{n+2}H_{xt} \\
&\quad + [\alpha^{n-1}]_{xt}H^{n+2}R_x + \alpha_x^{n-1}(n+2)H^{n+1}H_t R_x + \alpha_x^{n-1}H^{n+2}R_{xt} \\
&\quad + [\alpha^{n-1}]_t(n+2)H^{(n+1)}H_x^2 + \alpha^{n-1}(n+2)(n+1)H^n H_t H_x^2 \\
&\quad + \alpha^{n-1}(n+2)H^{n+1}2H_x H_{xt} \\
&\quad + [\alpha^{n-1}]_t(n+2)H^{n+1}H_x R_x \\
&\quad + \alpha^{n-1}(n+2)(n+1)H^n H_t H_x R_x \\
&\quad + \alpha^{n-1}(n+2)H^{n+1}H_{xt}R_x \\
&\quad + \alpha^{n-1}(n+2)H^{n+1}H_x R_{xt} \\
&\quad + [\alpha^{n-1}]_t H^{n+2}H_{xx} \\
&\quad + \alpha^{n-1}(n+2)H^{n+1}H_t H_{xx} \\
&\quad + \alpha^{n-1}H^{n+2}H_{xxt} \\
&\quad + [\alpha^{n-1}]_t H^{n+2}R_{xx} \\
&\quad + \alpha^{n-1}(n+2)H^{n+1}H_t R_{xx} \\
&\quad + \alpha^{n-1}H^{n+2}R_{xxt}
\end{aligned}
$$

Note that terms with a time derivative of bedrock such as $R_{xt}$ can be set to 0 since $R$ is assumed to be static in time. However, we keep the time derivatives for $R$ in the above equation for full generality in case a scenario is revisited where this does not hold. Next we derive $[\alpha^{n-1}]_t$:

$$
[\alpha^{n-1}]_t = \frac{n-1}{2}(S_x^2 + S_y^2)^{\frac{n-3}{2}}(2S_x S_{xt} + 2S_y S_{yt})
$$

Next we derive $[\alpha^{n-1}]_{tx}$:

$$
\begin{aligned}
[\alpha^{n-1}]_{tx} &= \frac{n-1}{2}[\frac{n-3}{2}(S_x^2 + S_y^2)^{\frac{n-5}{2}}(2S_x S_{xx} + 2S_y S_{yx})(2S_x S_{xt} + 2S_y S_{yt}) \\
&\quad + (S_x^2 + S_y^2)^{\frac{n-3}{2}}(2S_{yx}S_{yt} + 2S_y S_{ytx} + 2S_{xx}S_{xt} + 2S_x S_{xtx})]
\end{aligned}
$$

Next we derive $[\alpha^{n-1}]_{ty}$:

$$
\begin{aligned}
[\alpha^{n-1}]_{ty} =\ & \frac{n-1}{2}\Big[\frac{n-3}{2}(S_x^2+S_y^2)^{\frac{n-5}{2}}(2S_xS_{xy}+2S_yS_{yy})(2S_xS_{xt}+2S_yS_{yt}) \\
& + (S_x^2+S_y^2)^{\frac{n-3}{2}}(2S_{xy}S_{xt}+2S_xS_{xty}+2S_{yy}S_{yt}+2S_yS_{yty})\Big]
\end{aligned}
$$

Note that $S_{tx}=R_{tx}+H_{tx}=H_{tx}$ since $R$ is assumed to be fixed as a function of $t$. Note that the same argument holds for other derivatives of $S$ with respect to $t$. Next we derive $H_{tx},H_{txx},H_{ty},H_{tyy},H_{tyx}$:

$$
\begin{aligned}
H_{tx} &= -[\bar{u}H]_{xx}-[\bar{v}H]_{yx}+\dot{b}_{tx} \\
H_{txx} &= -[\bar{u}H]_{xxx}-[\bar{v}H]_{yxx}+\dot{b}_{txx} \\
H_{ty} &= -[\bar{u}H]_{xy}-[\bar{v}H]_{yy}+\dot{b}_{ty} \\
H_{tyy} &= -[\bar{u}H]_{xyy}-[\bar{v}H]_{yyy}+\dot{b}_{tyy} \\
H_{tyx} &= -[\bar{u}H]_{xxy}-[\bar{v}H]_{yyx}+\dot{b}_{tyx}
\end{aligned}
$$

Hence, these partial derivatives allow us to substitute purely spatial derivatives into the forward in time approximation for $H$. Without loss of generality, we use a central difference approximation for all spatial derivatives. Furthermore, we used $\Delta_t=.1$ years and $\Delta_x=\Delta_y=10^5$ m for the analysis in this paper. In total, 441 grid squares were modeled (i.e., 21 by 21) with the dome grid square at the origin. While a coarse grid was chosen for computational convenience, it is expected that numerical errors will go to zero as the grid width goes to zero, as is demonstrated both by Bueler et al. (2005) and Jarosch et al. (2013).

## Appendix B: Model fitting

In the following subsections, we go through the key details regarding Bayesian computation for the model used in this work. Assume $n$ total grid points are modeled, of which $m << n$ are observed. Let $X_j \in \mathbb{R}^n$ be the error correcting process at time $j$, $S_j \in \mathbb{R}^n$ be the latent glacier surface values at time $j$, $f(\theta,j) \in \mathbb{R}^n$ be shorthand for the output of the numerical solver at time point $j$, and $\epsilon_j$ be an **independent and identically distributed** (i.i.d) multivariate normal noise term at time $j$ with mean 0 and covariance matrix $\Sigma$. **(MVN stands for multivariate normal, and the first argument is the mean and the second is the covariance.)** Furthermore, assume that data is collected regularly at every $k_{th}$ time point, such that one observes $Y_k, Y_{2k}, ..., Y_{Nk} \in \mathbb{R}^m$, and the corresponding observation error $Z_k, Z_{2k}, ... Z_{Nk}$ is i.i.d $MVN(0,\sigma^2 I)$. For convenience, we denote $Nk$ as $T$. Finally, let $A \in \mathbb{R}^{m \times n}$ be a matrix which selects the grid squares of the latent process $S$ that are observed; that is, its rows are unit basis vectors corresponding to those indices that are observed.

### B1 Calculating the likelihood $p(Y_k, ..., Y_T | \theta)$

In this subsection, we derive both the likelihood of the observed data: $p(Y_k, ..., Y_T | \theta)$ **and an approximation to the likelihood**.

Though section 2.2 specifies the BHM in greater detail, the process and data levels of the BHM (i.e., conditioning on $\theta$) are concisely written as follows.

$$\begin{aligned} X_j &= X_{j-1} + \epsilon_j \\ S_j &= f(\theta, j) + X_j \\ Y_{ck} &= AS_{ck} + Z_{ck} \end{aligned}$$

Assume $j \in 1, 2, ...T$ and $c \in 1, 2, ..N$; hence there are $N$ **total spatial vectors** observed with a period of length $k$. Furthermore, $X_1$ is marginally $MVN(0, \Sigma)$. **That is, the process level vectors, $S_j$, are modeled conditional on the parameter level and the error correcting process. The data level vectors, $Y_{ck}$, are generated conditional on the process level $S_{ck}$. Throughout the following, we condition on $\theta$ being fixed.**

### B1.1  The exact likelihood

**Conditional on $\theta$, the distribution of $(Y_k, ..., Y_T)$, viewed as one long random vector, is multivariate normal. Also, conditional on $\theta$, the mean of $(Y_k, ..., Y_T)$ is $(Af(\theta, k), ..Af(\theta, T))$ because both $(X_k, ..., X_T)$ and $(Z_k, ..., Z_T)$ have mean 0. It suffices to thus derive the covariance matrix for $(Y_k, ..., Y_T)$ conditional on $\theta$. To do this, we note that** $Var(Y_{ck}) = Var(AS_{ck} + Z_{ck}) = Var(AS_{ck}) + Var(Z_{ck}) = [A(ck\Sigma)A^\intercal] + \sigma^2 I$. **Additionally, for $a < b$:**

$$\begin{aligned} Cov(Y_a, Y_b) &= Cov(AS_a + Z_a, AS_b + Z_b) \\ &= Cov(AS_a, AS_b) \\ &= Cov(A[f(\theta, a) + X_a], A[f(\theta, b) + X_b]) \\ &= Cov(AX_a, AX_b) \\ &= Var(AX_a) \\ &= [A(a\Sigma)A^\intercal] \end{aligned}$$

**Therefore, the covariance matrix for the observed data can be written as $M \otimes \Sigma + \sigma^2 I$, where $M_{ij} = kmin(i, j)$ and $M \in \mathbb{R}^{N \times N}$. This is a useful matrix representation because the inverse of $M$ is band-limited and sparse, for which there exist efficient computationally efficient linear algebraic routines (Rue, 2001).**

### B1.2  An approximation to the likelihood

The joint distribution $p(Y_k, ..., Y_T | \theta)$ can be written as $p(Y_k | \theta)p(Y_{2k} | Y_k, \theta)...p(Y_T | Y_k, .., Y_{(N-1)k}, \theta)$. Since we expect that the data level errors are quite small (on the order of 1m) in comparison to the overall surface elevation measurements (on the order of 1 km), we can approximate $p(S_{(c-1)k} | Y_k, .., Y_{(c-1)k}, \theta)$ with $p(S_{(c-1)k} | Y_{(c-1)k}, \theta)$. Consequently, $p(Y_{ck} | Y_k, .., Y_{(c-1)k}, \theta)$

will be close to $p(Y_{ck}|Y_{(c-1)k},\theta)$. From the above recursive relationship, we can write:

$$Y_{ck} \quad = \quad Y_{(c-1)k} + A[f(\theta,ck) - f(\theta,(c-1)k)] + Z_{ck} - Z_{(c-1)k} + \sum_{j=(c-1)k+1}^{ck} A\epsilon_j$$

**This expression motivates approximating $p(Y_{ck}|Y_k,..,Y_{(c-1)k},\theta)$ as MVN distribution with mean $Y_{(c-1)k} + A[f(\theta,ck) -$**
**$f(\theta,(c-1)k)]$ and covariance matrix $A(k\Sigma)A^\intercal + 2\sigma^2 I$. A similar expression shows that $p(Y_k)$ is multivariate normal**
**with mean $Af(\theta,k)$ and covariance matrix $A(k\Sigma)A^\intercal + \sigma^2 I$. Nonetheless, we must be clear: $p(Y_{ck}|Y_{(c-1)k},\theta)$ does not**
**exactly follow a MVN with mean $Y_{(c-1)k} + A[f(\theta,ck) - f(\theta,(c-1)k)]$ and covariance matrix $A(k\Sigma)A^\intercal + 2\sigma^2 I$; this**
**is because $Z_{(c-1)k}$ and $Y_{(c-1)k}$ are dependent. A simple example illustrating this approximation is presented in the**
**supplemental materials.**

## B2   Posterior computation

Posterior inference is accomplished with grid sampling (Gelman et al., 2013); this approach directly computes the posterior
distribution, $p(\theta|Y_k,...,Y_T)$ of the parameter, proportional to $p(Y_k,...,Y_T|\theta)p(\theta)$, on a grid of plausible values. The likelihood
is derived in the previous subsection. Parameters for the error correcting process are selected using knowledge elicited from
the studies of Bueler et al. (2005). **To verify the sensitivity of grid sampling to the grid width, three grid widths for $B$ are**
**considered: .25, .50, and 1, and the grid's range is from [1,70] (all in units of $10^{-25}\ s^{-1}Pa^{-3}$). The summary statistics**
**for generating $10^6$ posterior samples from more to less fine $(.25,.50,1)$ are given below:**

  – **Min: (5.25,5.00,6.00)**

  – **1st Quartile: (23.8,23.5,24.0)**

  – **Median: (27.0,26.5, 27.0)**

  – **Mean: (27.1,26.7,27.1)**

  – **3rd Quartile: (30.5,30.0,30.0)**

  – **Max: (51.50,49.0,51.0)**

**The similarity of summary statistics across grid widths indicates that the posterior samples are not very sensitive to**
**grid width; a grid width of .50 was used for the analyses within. Moreover, the posterior samples in this check were**
**generated for test case D (i.e., mass balance field producing a periodic solution to the SIA).**

## B3   Making spatio-temporal predictions of glacial surface elevation

In this section, we give details for how to make predictions under the proposed Bayesian model. Denote $S_{T_{end}} \in \mathbb{R}^n$ for
future glacier elevation values we want to make a prediction for at time point $T_{\text{end}}$. Our goal is to approximate the posterior

predictive distribution $p(S_{T_{\text{end}}}|Y_k,...Y_T)$. To make this computationally simple, our first assumption (as in the computation of the likelihood) is to suggest that $p(S_T|Y_k,...Y_T,\theta)$ is approximately equivalent to $p(S_T|Y_T,\theta)$. This is because relative to the overall glacier surface elevation values (an average of about 2000 m), the measurement errors are small, on the order of 1 m. Moreover, based on the model specified above, we know that $S_{T_{\text{end}}} = X_T + \sum_{j=T+1}^{T_{\text{end}}} \epsilon_j + f(\theta, T_{\text{end}})$. This suggests the following iterative procedure to generate a posterior sample for the prediction of $S_{T_{\text{end}}}$: for each independent sample $\theta_l$ from $p(\theta|Y_k,...,Y_T)$, generate a sample from a multivariate normal whose mean is 0 and covariance given by $(T_{\text{end}} - T)\Sigma$, add the sample to $f(\theta_l, T_{\text{end}})$, and then add this sum to a sample from $p(X_T|\theta = \theta_l, Y_T)$.

We must then determine how to sample from the distribution of $p(X_T|\theta = \theta_l, Y_T)$. Let $X_{\text{Tobs}} \in \mathbb{R}^m$ be a subvector of $X_T$ corresponding to the indices that are observed at the data level, and $X_{\text{Tpred}} \in \mathbb{R}^{n-m}$ be a subvector of $X_T$ corresponding to unobserved indices. The distribution for $p(X_{\text{Tobs}}|\theta, Y_T)$ is multivariate normal due to conjugacy. The precision, denoted by $Q_{\text{obs}}$, is $\sigma^{-2}I + [A(T\Sigma)A^{\mathsf{T}}]^{-1}$. The mean, denoted by $\mu_{\text{obs}}$, is $Q_{\text{obs}}^{-1}(\sigma^{-2}IY_T + [A(T\Sigma)A^{\mathsf{T}}]^{-1}Af(\theta,T)) - Af(\theta,T)$. $p(X_{\text{Tpred}}|X_{\text{Tobs}}, \theta, Y_T)$ is multivariate normal, whose mean and variance can be derived with the well-known conditional multivariate normal formula, as in Theorem 2.44 of Wasserman (2013). That is, the mean is $T\Sigma_{\text{pred,obs}}Q_{obs}$ and the variance is $T\Sigma_{\text{pred,pred}} - T\Sigma_{\text{pred,obs}}Q_{obs}T\Sigma_{\text{obs,pred}}$. Here, $\Sigma_{\text{pred,obs}}$ is the submatrix of $\Sigma$ that contains the rows of $\Sigma$ that correspond to the indices that are to be predicted, and the columns correspond to the indices which are observed. $\Sigma_{\text{obs,pred}}$ is analogously defined.

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
