# Peer review of "A Bayesian Hierarchical Model for Glacial Dynamics Based on the Shallow Ice Approximation and its Evaluation Using Analytical Solutions"

_The Cryosphere, 2017_

## Referee Comment (RC1) · L. Caron (Referee) · 17 Apr 2018

**I. General Comments**

The paper proposes to apply the Bayesian hierarchical model (BHM) framework to glacier modeling under the shallow ice approximation (SIA), using analytical solutions to the partial linear equation system. A unique aspect of the work is a sophisticated numerical error correction scheme through two dimensions of space and time, based on a statistical model. The authors conclude that their method is able to infer meaningful probability distributions for glacial parameters and predictions of the ice thickness, adequately accounting for the error originating from the numerical solver as well as

uncertainty in the parameters.

The problem is well-framed and the authors introduce it clearly through general explanations and context in sections 1 and 2, which greatly participates to the accessibility of the paper to non-specialists. The figures and tables are straightforward to interpret and informative, although in some cases expanded captions would be more useful if more information was reminded to the user given the length of the paper (see technical comments below).

The contribution is original and significant to advancing uncertainty quantification methods in the field of cryosphere science and seems promising in generating more applied follow-up works. On the other hand, however, I think the manuscript could be further improved, particularly by expanding the discussion section on the potential and limitations of the BHM for the broader community. I support the publication of the paper after minor revisions related to the following points.

II. Specific comments

1. The authors briefly mention in the summary and discussion that the method is applicable to broader problems in cryosphere science. Without going into further calculation, I believe expanding on that topic in the discussion would both provide better contextualization of the problem tackled here and increase the impact of the paper. What challenges do you expect for the cryosphere science community to apply BHM approach to the non-SIA regime, e.g. for fast-discharge ice streams, or to SIA problems without analytical solution (e.g. more realistic geometry)?

2. The authors manage well to point out the limitations introduced by simplifications in the physical problem and choices for the statistical distribution of errors. However, even after a few readings, it remains a bit difficult for me to tell what are the limits or downsides of the BHM approach itself, particularly on the resolution of parameters and state variables (e.g. ice thickness or velocity field).

- In the context of the SIA equations, can you say something about the relationship between the number of observations and the number of parameters? That is, how does the posterior evolve in the different cases with respect to the number of observations?

- Given the symmetry and choice of displaying only one quadrant in Figure 4, I wonder if the information (or uncertainty quantification) retrieved on the ice viscosity reflects that of 8 observations or that of $\sim$32. If one would compute a similar problem with a non-idealized glacier, how many observations would one need to obtain a similar posterior distribution for ice viscosity?

- Similarly, do the authors expect the spatial distribution of the observations to play a critical role in determining the posterior given the different sensitivity of the dome, margin and interior of the glacier?

3. Although the true value always remains within the confidence interval, there seems to be a tendency to under-predict the ice viscosity (as seen in Table 2) and over-predict the thickness (Figure 5). Is there any reason for that or is this purely the result of randomness?

4. In a non-linear PDE system, it is not guaranteed that the posterior is Gaussian or even symmetric distribution (even when propagating Gaussian errors). While the authors put a certain emphasis on the ice viscosity and basal sliding parameter, with respect to which the problem is linear, this linearity might not hold in general for every parameter or state variable one might want to keep track of. After all, a major appeal of Bayesian methods is that they require no assumption on the physics that are being solved, and are thus well suited to nonlinear problems. With that in mind, I believe using an accurate but more general terminology would be beneficial to future users of this work:

- p10 l18-20: "the .99 posterior credibility interval was computed by taking 3 standard deviations below and above the maximum a posteriori estimate (MAP) of the posterior samples." Even though these indicators are equal for a Gaussian (or any symmetric)

distribution, as a principle I would advise to refer to the mean or median instead of the maximum, as the former remain comparatively more adapted to characterize distributions even when they are not Gaussian. Perhaps the authors should also remind the reader that in a general (non-Gaussian) case, a distribution is best characterized by multiple indicators, e.g. quantiles as in Figure 5, and not just maximum and standard deviation.

- Throughout the manuscript the authors use interchangeably the phrases "3-Sigma" and ".99 Confidence" interval, as pointed out above. In a Gaussian distribution, the 3-sigma interval accounts for $\sim$0.9973 of the integral while the .99 interval represents $\sim$2.58-sigma, and clearly these are not the same. I think the authors should clarify and streamline this. It might otherwise introduce confusions and discrepancies in the exact numbers for readers that try to reproduce the results or compare them with a slightly different model setup (e.g. different geometry), especially if their method is based on numerical integration of the posterior.

- I recommend the authors to display the posterior distribution of $\mu$max, as a supplemental figure. Likewise, Figure 7 suggests non-symmetric probability distributions of the thickness originating from the error propagation, it might be beneficial to highlight the non-linearity by plotting these distributions in a similar way as Figure 6.

III. Technical comments -Figure 5: Outside of the whiskers, small circles are displayed, but the caption doesn't indicate what they are. If they are important, the authors should improve their visibility and add explanations related to them in the caption. If these are not meaningful on the other hand, the authors should remove them.

-Figures 5, 6, 7: when referring to test cases, remind the readers the specificity of these tests, e.g. "test case B (no mass balance or basal sliding)". This would lessen the need for cross-referencing.

-Table 2: The exponents of units are not displayed in superscript.

-Table 3: Is the dome error not calculated the same way as the margin and the interior? If so, I did not find any explanation in the text. If not, I suggest that the authors stream-line the column labels. Also, the authors should expand in the caption what RMSE stands for.

-Table 4: The authors should remind in the caption what the different symbols refer to.

I hope the authors will find this useful.

Lambert Caron

---

## Referee Comment (RC2) · Anonymous Referee #2 · 6 Jun 2018

Overall Review:

I believe this is an interesting, useful contribution and publishable with some revisions. Essentially, your computations assess the errors in using the numerical approximations for "f" using analytical solutions as a base line. That is, you generate "Y's" with analytical solutions but then forget about that and use numerical approximations in the BHM. "error" is then viewed as differences between Bayesian results and the analytical "truth". This is valuable work, though as you make clear, it doesn't make any assurances when the analytical model is "replaced by nature" in producing data. You also considered several cases, but I do think that your paper would be strengthened if you

also studied the impact sampling plans and sample sizes (ie. What if "every other observation (in time) was removed? This is also critical in judging the impacts of your approximations used in computations (see the next paragraph).

My first concern is correctness of all contributions. Errors can occur when manipulating equations rather than probability distributions. I think yours turned out right, but all conditioning assumptions are not clear. Consider Appendix B1 beginning at the bottom of p. 20. The "overall model" as written at the top of p. 21 is quite brief and does not include probability assumptions. I sense that you understand the key issues based on the sentence in lines 16-17, p. 21. Namely, equations like Y = m(variables) + error are code for "the conditional distribution of Y given "variables" and the mean of "error" = 0 and some variance of "error" has conditional mean m and conditional variance equal to the variance of "error". The assertion that all "errors" in you models have mean zero seems to be missing, but more importantly, when you do the manipulation leading to line 14, you must have assumed both models for Yck and Y(c-1)k are conditioned on the same quantities so you can simply subtract their conditional means, etc. Further, simply taking differences of Yck and Y(c-1)k is based on their joint distribution, so cavalierly moving Y(c-1)k to the left hand side and claiming you're now looking as the distribution of Yck given Y(c-1)k and the other variables. That requires a probability computation (moving from joint to a conditional distribution) in general. Fortunately, it is common that the algebraic versions can actually be proven to be correct probabilistically for "linear manipulations", but in complicated settings, this needs to be checked (based on my quick check, I think you're OK but think you should check as well). This all relates to my suggestion that your model isn't simply lines 2-4, p. 21. What are the conditional distributions assumption (the Z's are independent etc.)? One more related issue involves discussion of inference for X's. In a sense, you should be careful in posterior inferences about both S and X simultaneously, given q. (they are simply linear functions of each other). Again, I think you're OK but it merits your attention.

I think the approximations you used on p. 21 are reasonable, but a bit more defense

would be good. Further, I'm not comfortable with the way you needed all the approximations so that you could use grid sampling to claim genuine posterior inference. I think that you could ski p the approximations and did a full MCMC approach, it wouldn't be as easy as what you did but it's not that much harder. I think you should at least try some MCMC to confirm your computations and approximations. Further, what is the dependence of the value of your approximations on f. Surely you need to answer this if you plan to suggest operational use of you programs as you suggest you will do in the future.

Other Notes:

(1) The model for X is an explosive autoregression and hence you have built-in a limitation. A non-explosive model could be $X_j = r X_{j-1} +$ error where $0 < r < 1$. If you make r a parameter and let the data tell you about r, you may be able to predict further in the future if the data suggests r can be much smaller than 1.

(2) I think you missed emphasizing a crucial (and related) contribution of Berliner et al (2008). Namely they also treat model error through their "corrector process" and this should be mentioned.

(3) As a minor point, you should include at least one reference to

Berliner, L.M. 1996. Hierarchical Bayesian time series models. In Hanson, K. and R. Silver, eds. Maximum entropy and Bayesian methods. Dordrecht, etc., Kluwer Academic Publishers, 15–22.

The references by Wikle and Cressie both reference it but you should too since it urges the "data model, process model, parameter model" view. Also, since that paradigm is so key in your paper, I think you should break out the formula in line 22, p. 2 as a separate line for emphasis.

---

## Author Comment (AC1) · 20 Jun 2018

We are very pleased by the thorough, incisive, and constructive comments from the referees, which we believe have greatly improved the quality of the revised paper. Please see our responses to all of the referee comments, followed by a pointer to the modifications in the revised text. Please note that we have bolded the manuscript changes as to be able to easily identify them. Also, note that to mitigate redundancy, we have referenced the modifications made to the paper in the attached revision (TC_Author_Reply_revision_only.pdf in the supplement) instead of copying the modifications into this document. Kindly refer to TC_Author_Reply_revision_only.pdf while

reading the following comments.

Response to Dr. Lambert Caron:

"The figures and tables are straightforward to interpret and informative, although in some cases expanded captions would be more useful if more information was re-minded to the user given the length of the paper (see technical comments below)."

We appreciate the suggestion to make the captions more informative. Accordingly, we have modified the specific captions referenced in the technical comments, and we have also modified all of the additional captions as well.

"1. The authors briefly mention in the summary and discussion that the method is applicable to broader problems in cryosphere science. Without going into further cal-culation, I believe expanding on that topic in the discussion would both provide better contextualization of the problem tackled here and increase the impact of the paper. What challenges do you expect for the cryosphere science community to apply BHM approach to the non-SIA regime, e.g. for fast-discharge ice streams, or to SIA problems without analytical solution (e.g. more realistic geometry)?"

We appreciate here the call to discuss the generality of this approach with respect to problems in cryosphere science. Essentially, the same BHM can be used for other cryosphere science problems by swapping out the numerical solver for the SIA with a numerical solver to a different set of dynamics. The biggest challenge with this is satisfactorily modeling the numerical errors of this solver in a general way, discussed in the following paragraph.

Changes in revision: - Paragraph starting on line 11, page 5.

"2. The authors manage well to point out the limitations introduced by simplifications in the physical problem and choices for the statistical distribution of errors. However, even after a few readings, it remains a bit difficult for me to tell what are the limits or downsides of the BHM approach itself, particularly on the resolution of parameters and
state variables (e.g. ice thickness or velocity field)."

The BHM approach is not infallible, and the biggest difficulty is in actually fitting a BHM given data (despite that coming up with a BHM may not be too difficult). In our test cases, the number of physical parameters is small (1 or 2), so model fitting is not computationally difficult. However, when the number of physical parameters becomes larger (e.g., a basal sliding field with a parameter for every spatial location), posterior computation will become inefficient, and more sophisticated approaches will be needed (for which there is a large battery of tools). Besides a large number of physical parameters, another potential difficulty in utilizing BHMs that incorporate physical dynamics via a numerical solver is that the numerical solver can also be computationally onerous, so that posterior computation is very inefficient. While this is not a hindrance in the examples studied in this paper, in general this can be problematic. We have discussed this in the following paragraph.

Changes in revision: - Paragraph starting on line 17, page 3.

"- In the context of the SIA equations, can you say something about the relationship between the number of observations and the number of parameters? That is, how does the posterior evolve in the different cases with respect to the number of observations?"

The work of Brynjarsdottir and O'Hagan (2014) gives us an indication of how the posterior for physical parameters will evolve with more observations. In their work, they show that in a simple physical system with only a single parameter, some uncertainty in the posterior distribution for the physical parameter won't go away even as more data is collected. While this is attributed to improper modeling of the discrepancy between the output of a computer simulator and actual physical process values, and we have taken care in doing this, it is plausible that a similar phenomenon would occur in the BHM; that is, confounding between an error correcting process and the posterior of physical parameters results in posterior uncertainty never going away completely. In concordance with both this comment and the second referee's comments, we have included
posterior distributions for ice viscosity where the sampling period varies from once every 10 years, once every 5 years, once every year, and twice a year (as originally done to be consistent with summer and winter mass balance measurements per year). What we found is displayed in Figure 9; the general trend is that as more data is collected, the posterior becomes less biased but more diffuse. Thus, having posterior uncertainty that doesn't go away as more data is collected appears consistent with Brynjarsdottir and O'Hagan (2014).

Changes in revision: - Paragraph starting on line 8, page 13. - Addition of Figure 9.

"- Given the symmetry and choice of displaying only one quadrant in Figure 4, I wonder if the information (or uncertainty quantification) retrieved on the ice viscosity reflects that of 8 observations or that of 32. If one would compute a similar problem with a non-idealized glacier, how many observations would one need to obtain a similar posterior distribution for ice viscosity?"

To clarify, Figure 4 displays test locations for predictions, but not the locations where data are collected. The locations where surface elevation data are collected are distributed across the glacier at 25 locations as delineated in Section 3.2; to clarify this we have included a map marking the locations of these measuring sites. We will update the manuscript to clarify this in the caption. A similar number of locations would be adequate in a non idealized glacier, so long as the locations included points of steep changes in glacial thickness (e.g., valleys and peaks), since it appears that numerical errors are largest at such locations (i.e., the dome and margin).

Changes in revision: - Lines 17 and 18 of page 9. - Addition of Figure 4. - Caption of Figure 5.

"- Similarly, do the authors expect the spatial distribution of the observations to play a critical role in determining the posterior given the different sensitivity of the dome, margin and interior of the glacier?"

The biggest numerical errors occur where there are sharp changes in glacial thickness (i.e., peaks and valleys), such as the dome and the margin in the idealized glacier studied in Bueler (2005) and in this work. It is crucial, therefore, to ensure that such locations are sampled. Qualititatively, it is suggested that locations where there is a rapid change in glacier thickness ought to be sampled to ensure that numerical errors are adequately represented; the locations sampled in our simulation study include both the dome and locations close to the margin. The number of samples needed in general will then depend on the number of peaks and valleys in the glacier.

" 3. Although the true value always remains within the confidence interval, there seems to be a tendency to under-predict the ice viscosity (as seen in Table 2) and over-predict the thickness (Figure 5). Is there any reason for that or is this purely the result of randomness?"

A very similar phenomenon has been documented in the work of Brynjarsdottir and O'Hagan (2014); in their work, it is noted that good prior information must be encoded into model discrepancy (essentially what we have termed an error correcting process) and physical parameters in order to get a less biased posterior distribution for both physical parameters and predictions. A similar phenomenon can be demonstrated in our BHM. In particular, if we consider a scenario where we ignore the prior information regarding different scales of numerical errors between the interior, dome, and margin of the glacier, the bias of the posterior distribution for physical parameters is more pronounced. So while bias of the posterior of physical parameters exists in our simulation studies, the prior information we have used appears to have helped reduce this bias, consistent with the findings of Brynjarsdottir and O'Hagan (2014). We have revised the manuscript to include an example illustrating this point in the results section.

Changes in revision: - Paragraph starting on line 27, page 12. - Addition of Figure 8.

"4. In a non-linear PDE system, it is not guaranteed that the posterior is Gaussian or even symmetric distribution (even when propagating Gaussian errors). While the

authors put a certain emphasis on the ice viscosity and basal sliding parameter, with respect to which the problem is linear, this linearity might not hold in general for every parameter or state variable one might want to keep track of. After all, a major appeal of Bayesian methods is that they require no assumption on the physics that are being solved, and are thus well suited to nonlinear problems."

As a minor point of clarification for readers, the PDE on line 11 page 6 is non-linear in H, glacial thickness, since it involves powers of H. However, I suspect the use of linear here refers to the fact that B and $C_{0gamma}$ appear as constants (i.e., not functions thereof) in these equations.

"With that in mind, I believe using an accurate but more general terminology would be beneficial to future users of this work:

- p10 l18-20: the .99 posterior credibility interval was computed by taking 3 standard deviations below and above the maximum a posteriori estimate (MAP) of the posterior samples. Even though these indicators are equal for a Gaussian (or any symmetric) distribution, as a principle I would advise to refer to the mean or median instead of the maximum, as the former remain comparatively more adapted to characterize distributions even when they are not Gaussian. Perhaps the authors should also remind the reader that in a general (non-Gaussian) case, a distribution is best characterized by multiple indicators, e.g. quantiles as in Figure 5, and not just maximum and standard deviation."

The MAP was used to be consistent with the previous related work in Brinkerhoff et al. (2016), but we have used the mean instead of the MAP in the revision (the results are essentially the same). It should be noted that constructing a credibility interval in this way (mean +/- 3 sd of posterior samples) does not necessitate that the posterior distribution is Gaussian.

Changes in revision: - Sentence starting on line 3, page 12. - Sentences starting on line 15, page 12. - Sentence starting on line 23, page 12. - Change of .99 credibility

interval to 3-sd credibility interval in Table 2.

" - Throughout the manuscript the authors use interchangeably the phrases "3-Sigma" and ".99 Confidence" interval, as pointed out above. In a Gaussian distribution, the 3-sigma interval accounts for 0.9973 of the integral while the .99 interval represents 2.58-sigma, and clearly these are not the same.

I think the authors should clarify and streamline this. It might otherwise introduce confusions and discrepancies in the exact numbers for readers that try to reproduce the results or compare them with a slightly different model setup (e.g. different geometry), especially if their method is based on numerical integration of the posterior."

Thank you for pointing out the potential confusion this can cause. To be consistent, we have updated the terminology to be '3-sd credibility interval' (again, constructed with mean +/- 3 sd of posterior samples).

Also it should be noted that credibility interval refers to an interval derived from a posterior distribution, which is distinct from a confidence interval. The latter has a particular frequentist coverage probability.

The changes in the revision are the same as above, that is:

- Sentence starting on line 3, page 12. - Sentences starting on line 15, page 12. - Sentence starting on line 23, page 12. - Change of .99 credibility interval to 3-sd credibility interval in Table 2.

"- I recommend the authors to display the posterior distribution of $\mu$max, as a supplemental figure. Likewise, Figure 7 suggests non-symmetric probability distributions of the thickness originating from the error propagation, it might be beneficial to highlight the non-linearity by plotting these distributions in a similar way as Figure 6."

We have included a posterior plot of mu_max in the supplemental materials. In our humble collective opinion, individual predictive density plots do not appear to convey more information than Figure 6, so we have opted not to include these.

[Figure]

"III. Technical comments -Figure 5: Outside of the whiskers, small circles are displayed, but the caption doesn't indicate what they are. If they are important, the authors should improve their visibility and add explanations related to them in the caption. If these are not meaningful on the other hand, the authors should remove them."

It is typical for box and whisker plots to display outliers, defined as more than 1.5 times the interquartile range beyond the first and third quartiles; these outliers are displayed as circles. Agreeably, it is important for us to be clear about this, so we have included a note in the caption.

Changes in revision: - Last two sentences of the caption in Figure 6.

"-Figures 5, 6, 7: when referring to test cases, remind the readers the specificity of these tests, e.g. "test case B (no mass balance or basal sliding)". This would lessen the need for cross-referencing."

Thank you for this suggestion, which we have taken heed of in the revision. Changes in revision: - Caption of Figure 6. - Caption of Figure 7. - Caption of Figure 8. - Caption of Figure 9. - Caption of Figure 10.

"-Table 2: The exponents of units are not displayed in superscript."

Thank you for spotting this, which we have corrected.

Changes in revision: - Last row of Table 2.

"-Table 3: Is the dome error not calculated the same way as the margin and the interior? If so, I did not find any explanation in the text. If not, I suggest that the authors streamline the column labels. Also, the authors should expand in the caption what RMSE stands for."

The dome error is calculated in the same way as the margin and interior, but since there is only a single dome observation, RMSE, which stands for root mean squared error, is just the absolute difference between the actual and predicted. Nonetheless, to remain

consistent we have changed dome error to be RMSE. Thank you for pointing out that we ought to include what RMSE stands for, which we have revised in the manuscript.

Changes in revision: - Header of Table 3. - Caption of Table 3.

"-Table 4: The authors should remind in the caption what the different symbols refer to."

We certainly agree and have revised the manuscript accordingly. Changes in revision: - Caption of Table 4.

"I hope the authors will find this useful."

These comments have been extremely valuable for improving the manuscript; the referee's time and effort are appreciated.

Response to anonymous referee 2:

"I believe this is an interesting, useful contribution and publishable with some revisions. Essentially, your computations assess the errors in using the numerical approximations for "f" using analytical solutions as a base line. That is, you generate "Y's" with analytical solutions but then forget about that and use numerical approximations in the BHM. "error" is then viewed as differences between Bayesian results and the analytical "truth". This is valuable work, though as you make clear, it doesn't make any assurances when the analytical model is "replaced by nature" in producing data. You also considered several cases, but I do think that your paper would be strengthened if you also studied the impact sampling plans and sample sizes (ie. What if "every other observation (in time) was removed? This is also critical in judging the impacts of your approximations used in computations (see the next paragraph)."

In order to consider the impact of sampling plans, we have conducted an additional set of simulation studies where the period of observations varies: once every 10 years, once every 5 years, once every year, and twice a year (as originally conducted); however, please note that we chose two measurements per year to model how the data

set from the University of Iceland was collected – namely, a set of measurements for winter and summer mass balance.

Changes in revision: - Paragraph starting on line 8, page 13. - Addition of Figure 9.

"My first concern is correctness of all contributions. Errors can occur when manipulating equations rather than probability distributions. I think yours turned out right, but all conditioning assumptions are not clear. Consider Appendix B1 beginning at the bottom of p. 20. The "overall model" as written at the top of p. 21 is quite brief and does not include probability assumptions. I sense that you understand the key issues based on the sentence in lines 16-17, p. 21. Namely, equations like Y = m(variables) + error are code for "the conditional distribution of Y given "variables" and the mean of "error" = 0 and some variance of "error" has conditional mean m and conditional variance equal to the variance of "error"."

Thank you for pointing out some places where the probabilistic assumptions of the BHM can be made clearer. The probabilistic assumptions are specified in Section 2.2 and in the first paragraph of Appendix B. In particular, please note that in the first paragraph of Appendix B line 21 it is stated: "let epsilon_j be an independent and identically distributed MVN(0,$\Sigma$) noise term at time j", and also in Appendix B line 21 it is stated: "the corresponding observation error Zk,Z2k,...ZNk is i.i.d MV N(0,$\sigma$2 I)". The use of acronyms may make these lines unwieldy to parse, so we have revised them (that is, i.i.d is independent and identically distributed, and MVN(0,$\Sigma$) is multivariate normal with mean 0 and covariance $\Sigma$).

Additionally, we have stressed in the revision that we are conditioning on theta when computing the likelihood.

Changes in revision: - Lines 21-23, page 24 - Lines 28-30, page 24. - Lines 5-8, page 25.

"The assertion that all "errors" in you models have mean zero seems to be missing,

but more importantly, when you do the manipulation leading to line 14, you must have assumed both models for Yck and Y(c-1)k are conditioned on the same quantities so you can simply subtract their conditional means, etc. Further, simply taking differences of Yck and Y(c-1)k is based on their joint distribution, so cavalierly moving Y(c-1)k to the left hand side and claiming you're now looking as the distribution of Yck given Y(c-1)k and the other variables. That requires a probability computation (moving from joint to a conditional distribution) in general. Fortunately, it is common that the algebraic versions can actually be proven to be correct probabilistically for "linear manipulations", but in complicated settings, this needs to be checked (based on my quick check, I think you're OK but think you should check as well). This all relates to my suggestion that your model isn't simply lines 2-4, p. 21. What are the conditional distributions assumption (the Z's are independent etc.)?"

Thank you for the call to clarify the arguments made in this section. Regarding errors having mean zero, please note that this is stated in the first paragraph of Appendix B. Regarding the remaining comments, we have taken a number of actions.

First, we have included a derivation of the complete likelihood without using any approximations before going into the approximation.

Second, we have clarified an important point you have raised, which is that the expression on line 14 page 21 cannot be used to claim that the distribution of p(Y_ck|Y_(c-1)k) is *exactly* a MVN distribution with mean A[f $(\theta,$ ck)$-$ f$(\theta,$(c$-$1)k)] and covariance matrix A(k$\Sigma$)A $+2\sigma$ 2 I; this is because Y_(c-1)k and Z_(c-1)k are not independent. This expression, rather, motivates approximating p(Y_ck|Y_(c-1)k) with said MVN distribution. We have rewritten the text in this portion to be more clear about this point.

Third, we have included a simple example illustrating why this approximation is reasonable under the assumption that the output of the numerical solver is much larger in magnitude compared to the measurement error.

Changes in revision: - Subsection B1.1 on page 25. - Paragraph starting on line 4 on

page 26. - Supplementary note with an illustrative example regarding the approximation used.

"One more related issue involves discussion of inference for X's. In a sense, you should be careful in posterior inferences about both S and X simultaneously, given q. (they are simply linear functions of each other). Again, I think you're OK but it merits your attention."

Indeed, as per the equations on the top of p. 21 (of the original submission, but page 25 of the revision), *conditioning on theta*, S is just X shifted by the output of the numerical solver. It is very important to stress that this is conditional on theta (the output of the numerical solver is fixed conditioning on theta).

"I think the approximations you used on p. 21 are reasonable, but a bit more defense would be good."

Please see the aforementioned simple example in the supplemental materials regarding this issue.

"Further, I'm not comfortable with the way you needed all the approximations so that you could use grid sampling to claim genuine posterior inference. I think that you could skip the approximations and did a full MCMC approach, it wouldn't be as easy as what you did but it's not that much harder. I think you should at least try some MCMC to confirm your computations and approximations."

It should be made clear that the approximation for the likelihood was not needed to compute the posterior on a grid. Many (though not all) MCMC algorithms require (log) likelihood evaluations as well, and a computationally inefficient (log) likelihood will mitigate their performance. Since we are working with only 1 to 2 physical parameters in test cases (B for the first three test cases, and B,mu_max in the last test case), computing the posterior on a grid ought to perform just as good as an MCMC approach, provided a sufficiently fine grid is chosen. Moreover, it is important to point out that the

output of MCMC samples can be flawed (e.g., only one mode of a complex posterior is explored) and the samples thus may not actually reflect the posterior distribution.

This being said, we admit that we could have done a better job of checking the sensitivity of the grid sampling approach to the particular grid we had used. To check that the posterior for physical parameters is not sensitive to using a grid, we have computed the posterior on various grid widths for comparison, and have quoted summary statistics for posterior samples for comparison. The summary statistics are very close, indicating that the choice of grid width we had used did not distort or severely misrepresent the posterior distribution.

Changes in revision: - Section B2 from line 14 onwards.

"Further, what is the dependence of the value of your approximations on f. Surely you need to answer this if you plan to suggest operational use of you programs as you suggest you will do in the future."

We appreciate here the call to clarify the requirement for applying the likelihood approximation. The requirement for applying the likelihood approximation is that the values of S, and consequently of f, are much less than the measurement error – this holds in the scenario of this paper because the values of f are on the order of one kilometer, whereas the measurement error is one meter. Please see the aforementioned simple illustrative example in the supplemental figure for justification on this point. However, the point raised about applying the code to future scenarios is certainly valid, since these conditions aren't always going to hold. We are currently in development of a way to efficiently calculate the log-likelihood in regimes where this does not hold (such as in other cryosphere problems with a poorer signal to noise ratio) and will include this functionality in the package mentioned.

Changes in revision: - Supplementary note with an illustrative example regarding the approximation used.

"Other Notes: (1) The model for X is an explosive autoregression and hence you have built-in a limitation. A non-explosive model could be $X_j = r X_{j-1} + $ error where $0 < r < 1$. If you make r a parameter and let the data tell you about r, you may be able to predict further in the future if the data suggests r can be much smaller than 1."

We agree that for more general models, it would be good to learn the parameter r directly from the data. However, for the time scales for the analysis used in this work, and based on the evidence from Figure 10, using $r = 1$ seemed to work out adequately. Another complication to consider is that learning an additional parameter along with the physical parameters can increase computational difficulties of posterior computation, though this is something we ought to more thoroughly investigate for future extensions of this model.

"2) I think you missed emphasizing a crucial (and related) contribution of Berliner et al (2008). Namely they also treat model error through their "corrector process" and this should be mentioned."

We appreciate the call to highlight this important contribution of Berliner et al. (2008) and have accordingly updated the text. Please note that the error correcting process in that work accounts for setting basal shear stress to driving stress, a simplification. Somewhat orthogonally, the error correcting process in this paper solely accounts for numerical errors due to an imperfect numerical solver, though it is still important to consider the fact that stress terms are not physically perfect as well.

Changes in revision: - Lines 7-8 on page 4.

"(3) As a minor point, you should include at least one reference to Berliner, L.M. 1996. Hierarchical Bayesian time series models. In Hanson, K. and R. Silver, eds. Maximum entropy and Bayesian methods. Dordrecht, etc., Kluwer Academic Publishers, 15–22. The references by Wikle and Cressie both reference it but you should too since it urges the "data model, process model, parameter model" view."

Thank you for pointing out this additional reference, which we have added to the manuscript.

Changes in revision: - Line 3 on page 3.

"Also, since that paradigm is so key in your paper, I think you should break out the formula in line 22, p. 2 as a separate line for emphasis."

Though there is no formula on line 22, page 2, we have broken out the formula on line 20, page 2 as suggested.

To reiterate, the comments from the reviewers are greatly appreciated, and we believe they have helped us significantly improve the quality of this work.

Additional materials: We have shared R scripts written for this paper in the supplemental materials, in case that they may be helpful for the community. As such, we have included scripts to: compute the analytical solutions in test cases B-E, run the finite difference method for test cases B-E, generate the simulations based on analytical solutions in test cases B-E.

Sincerely, Giri Gopalan and co-authors

Please also note the supplement to this comment:
https://www.the-cryosphere-discuss.net/tc-2017-275/tc-2017-275-AC1-supplement.zip